# Regional to global distributions, trends, and drivers of biogenic volatile organic compound emission from 2001 to 2020

Hao Wang[1], Xiaohong Liu[2], Chenglai Wu[1], and Guangxing Lin[1]

[1]International Center for Climate and Environment Sciences, Institute of Atmospheric Physics, Chinese Academy of Sciences, Beijing, China
[2]Department of Atmospheric Sciences, Texas A&M University, College Station, Texas, USA

*Correspondence to*: Xiaohong Liu (xiaohong.liu@tamu.edu)

**Abstract.** Biogenic volatile organic compounds (BVOCs) are important precursors to ozone and secondary organic aerosols in the atmosphere, affecting air quality, clouds and climate. However, the trend of BVOC emissions and driving factors for the emission changes in different geographic regions over the past two decades has remained unclear. Here, regional to global changes in BVOC emissions during 2001-2020 are simulated using the latest Model of Emission of Gases and Aerosols from Nature (MEGANv3.2) with the input of time-varying satellite-retrieved vegetation and reanalysis meteorology data. Comparison of model simulations with the site observations shows that the model can reasonably reproduce the magnitude of isoprene and monoterpene emission fluxes. The spatial distribution of the modeled isoprene emissions is generally comparable to the satellite retrievals. The estimated annual average global BVOC emissions are 835.4 $Tg\,yr^{-1}$ with the emissions from isoprene, monoterpenes, sesquiterpenes, and other BVOC comprised of 347.7, 184.8, 23.3, and 279.6 $Tg\,yr^{-1}$, respectively. We find that the decrease in global isoprene emissions ($-0.07\%\ yr^{-1}$) caused by increase in $CO_2$ concentrations ($-0.20\%\ yr^{-1}$) is stronger than that caused by changes in vegetation ($-0.03\%\ yr^{-1}$) and meteorological factors ($0.15\%\ yr^{-1}$). However, regional disparities are large. Isoprene emissions increase significantly in Europe, East Asia, and South Asia ($0.37-0.66\%\ yr^{-1}$). The increasing trend is contributed by half from increased leaf area index (LAI) (maximum over 0.02 $m^2\,m^{-2}\,yr^{-1}$) and tree cover. Changes in meteorological factors contribute to another half, with elevated temperature dominating in Europe and increased soil moisture dominating in East and South Asia. In contrast, in South America and Southeast Asia, shifts in vegetation type associated with the BVOC emission capacity, which partly results from the deforestation and agricultural expansion, decrease the BVOC emission and offset nearly half of the emission increase caused by changes in meteorological factors. Overall, isoprene emission increases by $0.35\%\ yr^{-1}$ and $0.25\%\ yr^{-1}$ in South America and Southeast Asia, respectively. In Central Africa, a decrease in temperature dominates the negative emission trend ($-0.74\%\ yr^{-1}$). Global monoterpene emissions show a significantly increasing trend ($0.34\%\ yr^{-1}$, $0.6\ Tg\,yr^{-1}$) compared to that of isoprene ($-0.07\%\ yr^{-1}$, $-0.2\ Tg\,yr^{-1}$), especially in strong greening hotspots. This is mainly because the monoterpene emissions are more sensitive to changes in LAI and are not subject to the inhibition effect of $CO_2$. The findings highlight the important roles of vegetation cover and biomass, temperature, and soil moisture in modulating the temporal variations of global BVOC emissions in past two decades.

## 1 Introduction

Emissions of biogenic volatile organic compounds (BVOCs) from the terrestrial vegetation play a pivotal role in atmospheric chemistry and climate due to their large quantity (~1000 Tg yr$^{-1}$) and high reactivity (Guenther et al., 1995, 2012). Isoprene and monoterpenes (e.g., α-pinene, β-pinene, limonene) are the most prevalent BVOC species, and other species include sesquiterpenes, methanol, ethanol, etc. These BVOCs account for about 90% of total non-methane volatile organic compounds (NMVOCs) in the atmosphere (Guenther et al., 2006), which are important precursors for troposphere ozone and secondary organic aerosols (SOA) through atmospheric oxidation processes, and thus influence air pollution, clouds and Earth's radiative budget. However, BVOC emissions remain highly uncertain as they depend on a diversity of factors and it is still unclear about the relative importance of different factors.

The BVOC emissions are determined by many environmental factors such as vegetation, meteorology, and carbon dioxide (CO$_2$) concentrations. The impact of vegetation on BVOC emissions is primarily reflected in vegetation types (e.g., forest, grassland), tree species, and vegetation biomass density (e.g., land cover, leaf area index). Previous studies have demonstrated that different vegetation types and tree species affect BVOC emissions dramatically (Lathière et al., 2006; Stavrakou et al., 2014; Sindelarova et al., 2022). For meteorological parameters, especially temperature, light, and soil moisture, it was observed that elevated temperature, stronger radiation, and wetter soil significantly promote BVOC emissions (Rinne et al., 2002; Bai et al., 2016; Jiang et al., 2018). In contrast, elevated CO$_2$ can suppress the emissions of the major BVOC component (e.g., isoprene) (Heald et al.,2009; Wilkinson et al., 2009). The vegetation change has affected nearly half of the global land surface (Hurtt, 2011) with land cover change intensifying in recent decades, particularly in tropical and East Asia (Purves et al., 2004; Pacifico et al., 2012; Piao et al., 2015). Related studies pointed out that 1/3 of the global vegetation growth area has been greening since the 21st century, with the leaf area index (LAI) increasing by 2.3% per decade, of which China contributes nearly 1/4 (Chen et al., 2019). In addition, the surge in greenhouse gas emissions since the industrial revolution has led to significant global warming and meteorological changes. Thus, due to the large spatio-temporal variations in these factors, there may exist significant regional and global differences in BVOC emissions as well as emission trends.

Ground-based measurements can sample BVOC fluxes from leaf to canopy scale (Müller et al., 2010; Bai et al., 2016; Sarkar et al., 2020). However, the spatial and temporal coverages of such observations are limited and cannot be extended to represent BVOC emissions in a larger domain. The BVOC emission models, such as the Model of Emissions of Gases and Aerosols from Nature (MEGAN, Guenther et al., 1995, 2006, 2012) and the BVOC Photosynthesis-Dependent Scheme (PS_BVOC, Arneth et al., 2007; Unger et al., 2013), consider the main factors influencing BVOC emissions. Compared to the widely used MEGAN version 2.1, the recently released MEGAN version 3.2 has addressed several gaps in BVOC emission modeling (Guenther et al., 2020), including (i) more refined BVOC emission factors, where vegetation emission factors are calculated based on tree species rather than the original fixed plant functional type (PFT) emission factor and (ii) consideration

of environmental stress caused by extreme weather and air pollution. These models have been applied to simulate the BVOC emissions from regional to global scales, and to identify the impacts of various factors on the variations in BVOC emissions (Fu and Liao, 2012; Purves et al., 2014; Stavrakou et al., 2014; Chen et al., 2018; Wang et al., 2021; Li et al., 2022).

Previous studies have explored the long-term emission trends of BVOCs. Chen et al. (2018) paid attention to changes in vegetation from 2000 to 2015 on the global scale, and reported that the global total isoprene emission declined by only 1.5% for this period. Opacka et al. (2021) complemented the work of Chen et al. (2018) by incorporating different satellite-retrieved land cover datasets and pointed out that land cover changes from 2001 to 2016 mitigate isoprene emissions ranging from $-0.33\%$ to $-0.04\%$ $yr^{-1}$, while temperature and radiation changes enhance isoprene emissions by 0.94% $yr^{-1}$. Sindelarova et al. (2022) argued that although vegetation change from 2000 to 2019 exerted a small effect ($-0.11\%$ $yr^{-1}$) on the overall change in global isoprene emissions, the changes were significant in some hotspots. Purves et al. (2004) found that changes in vegetation in the Eastern United States from 1980 to 1990 could lead to an increase of BVOC emissions by about 17%. Fu and Liao (2012) analyzed the changes in BVOC emissions in China from 2001 to 2006 and found that the interannual variability of isoprene emissions was dominated by changes in the meteorological fields, while the variability of monoterpene emissions was more sensitive to changes in vegetation. Stavrakou et al. (2014) explored the factors influencing isoprene emissions in Asia from 1979 to 2012 and found that enhanced temperature and light led to a 0.52% $yr^{-1}$ increase in emissions in China, and oil palm expansion in Southeast Asia increased the isoprene emissions by more than 1% $yr^{-1}$. Chen et al. (2018) showed that from 2000 to 2015, afforestation caused a $5-10\%$ increase in isoprene emissions in Northeastern China and India, while deforestation led to about a 10% reduction in isoprene emissions in the Amazon basin, West Africa, and Southeast Asia. Wang et al. (2021) indicated that the greening of China from 2001-2016 led to a significant increase (up to 11.7%) in BVOC emissions, and the regional accumulated BVOC emissions in 2018 could be 26% higher than those in 2001, mainly due to the increase in vegetation cover and LAI (Li et al., 2022). However, these previous studies were conducted at various spatio-temporal scales using fixed vegetation or meteorological datasets for a given year, leading to difficulties in comparing their magnitude and even sign of the BVOC emission trends, and it is still unclear which factor dominates the BVOC emission trends in different hotspot regions.

This study provides a comprehensive analysis of BVOC emission trends from 2001 to 2020 on a regional to global scale using the latest BVOC emission model, MEGANv3.2, combined with time-varying satellite vegetation retrievals and meteorological reanalysis data. More importantly, this study further identifies the contribution of various driving factors to these trends. The findings of this research shed light on the importance of BVOC emissions in air quality and aerosol radiative forcing in hotspot regions. This paper is organized as follows. In Section 2, we introduce the MEGANv3.2 model and its input data, model experimental design, and BVOC observation data used in model evaluations. In Section 3, the spatio-temporal distributions and trends of BVOC emissions in different regions are analyzed. The contributions of various driving factors (i.e.,

vegetation, meteorology, and $CO_2$) are quantified. In Section 4, we discuss the uncertainty of the model results and the sources
of biases in the BVOC emission trends by comparing with previous studies. Conclusions of this study are given in Section 5.

## 2 Method and Data

### 2.1 MEGANv3.2

The MEGAN emission model has been widely used to simulate BVOC emissions at global and regional scales (Guenther et al., 2006, 2012). It has also been incorporated into various Earth system models and chemical transport models (Müller et
al., 2008; Li et al., 2013; Sindelarova et al., 2014; Messina et al., 2016; Bauwens et al., 2018; Chen et al., 2018). Here, the latest version of MEGANv3.2 is applied to estimate the BVOC emissions from 2001 to 2020. Compared to the earlier version MEGANv2.1 (Guenther et al., 2012), MEGANv3.2 estimates vegetation emission factors based on variable plant species measurements instead of on fixed plant functional type (PFT, Guenther et al., 2020). Specifically, while MEGANv2.1 uses a look-up table of emission factors for the 15 PFTs corresponding to the biological emission classes (see Table 2 in Guenther et
al. (2012)), MEGANv3.2 uses the so-called Emission Factor Processor, to estimate the landscape average emission factors, which are based on the following three databases: (1) Growth form datasets for four PFTs: tree, shrub, grass, and crops; (2) Ecotype datasets: composed of a mix of emission-specific tree species/grass associated with specific emission capacities; and (3) Updated tree species/grass datasets corresponding to the biogenic emission classes. These updates can distinguish the differences in vegetation emission factors in regions with the same PFT but with varying plant species. The new version also
considers the additional stress factors of emissions by using the simple threshold function, including high/low temperature, strong wind, and heavy $O_3$ pollution. Additionally, the number of BVOC components in MEGANv3.2 is expanded from the original 148 to over 200. MEGANv3.2 calculates the BVOC emissions rate (ER) as follows:

$$ER = EF \cdot EA \qquad (1)$$

where $EF$ and $EA$ represent the standard emission factor (i.e., $ER$ at "standard" conditions) and the nondimensional emission
activity factor, respectively.

The $EF$ map can be obtained by running MEGANv3.2 Emission Factor Processor, which combines growth form and ecotype data with plant species community and species emission factor datasets to generate the mean emission factor. A brief algorithm is shown below:

$$EF = EF_{tree} \cdot f_{tree} + EF_{shrub} \cdot f_{shrub} + EF_{grass} \cdot f_{grass} + EF_{crop} \cdot f_{crop} \qquad (2)$$

where $EF_{tree}$, $EF_{shrub}$, $EF_{grass}$, and $EF_{crop}$ represent the species emission factors for the four types of growth forms (i.e., PFTs), and $f$ is the fraction of the specific growth form in a model grid cell.

The emission activity factor considers the effect of various environmental factors and is calculated as

$$EA = \text{LAIv} \cdot \gamma_p \cdot \gamma_T \cdot \gamma_{HT} \cdot \gamma_{LT} \cdot \gamma_{SW} \cdot \gamma_{O_3} \cdot \gamma_A \cdot \gamma_{SM} \cdot \gamma_C \qquad (3)$$

where LAIv represents the leaf area index of vegetation covered surfaces and is obtained by dividing LAI with VCF (vegetation cover fraction). $\gamma_p$, $\gamma_T$, $\gamma_{HT}$, $\gamma_{LT}$, $\gamma_{SW}$, $\gamma_{O_3}$, $\gamma_A$, $\gamma_{SM}$, $\gamma_C$ represent the activity factors for downward shortwave radiation, 2-m air temperature, high temperature, low temperature, strong wind, $O_3$ pollution, leaf age, soil moisture, and $CO_2$ concentration, respectively. In MEGANv3.2 the increases in temperature, radiation, and soil moisture favor the BVOC emissions, while high/low temperature (>40 °C or <10 °C) and strong wind (>12 m s$^{-1}$) as well as heavy $O_3$ pollution and high $CO_2$ concentration suppress the BVOC emissions. Currently, the inhibition effect of $CO_2$ on BVOC emissions in the model is only available for isoprene. In this study, we consider the effect of all above activity factors on BVOC emissions except for the $O_3$ pollution factor (i.e., $\gamma_{O_3} = 1$). More details of these algorithms are described in Guenther et al. (2006, 2012, 2020).

## 2.2 Data

### 2.2.1 Vegetation datasets

The vegetation parameters driving MEGANv3.2 include LAI, VCF, and PFT. In this study, the Moderate-resolution Imaging Spectroradiometer (MODIS) vegetation retrievals from 2001 to 2020 were used. LAI data was obtained from Yuan et al. (2011), which improved the MODIS version 6 product MCD15A2H (Myneni et al., 2015) with a temporal resolution of 8 days and a spatial resolution of 0.5°×0.5°. The LAIv calculated in MEGANv3.2 is defined as LAI divided by VCF, representing the leaf area index per unit vegetation area. The VCF was from the yearly MODIS MOD44B version 6 dataset (DiMiceli et al., 2015), which contains three ground cover components (i.e., tree cover, non-tree cover, and bare soil cover). We summed the first two observed variables as vegetation cover. The raw VCF product (250 m pixel size, sinusoidal grid) was further converted to a 0.5°×0.5° latitude/longitude grid by a conservation interpolation before used in MEGANv3.2. The PFT was obtained from the yearly MODIS MCD12C1 product with a spatial resolution of 0.05° (Friedl and Sulla-Menashe, 2015). To calculate the growth form fractions used by the Emission Factor Processor in Equation 2, the selected 17 MODIS IGBP (International Geosphere Biosphere Programme) global vegetation classification types from above MODIS PFT product were mapped to four main PFT classification types (i.e., tree, shrub, grass, and crop) in MEGANv3.2 based on methods from Sulla-Menashe and Friedl (2018). The ecotype dataset mentioned in Section 2.1 is based on satellite imagery and ground surveys and comes from the MEGAN model development group (https://bai.ess.uci.edu/megan/data-and-code/growth-form-and-ecotypes, last access: 21 Nov 2022) The reprocessed datasets were conservatively interpolated to a spatial resolution of 0.5°×0.5° as model inputs.

### 2.2.2 Meteorological datasets

The meteorological parameters driving the MEGANv3.2 model were from the Modern-Era Retrospective analysis for Research and Applications, Version 2 (MERRA-2) (Gelaro et al., 2017). To reduce the biases in the simulated radiation flux trends from aerosols, MERRA-2 has assimilated aerosol optical depths from space-based observations in the long-term and considered the interaction of aerosols with the climate system. The selected variables used in MEGANv3.2 include 2 m

temperature, surface downward shortwave radiation, surface soil moisture, water vapor mixing ratio, 10 m wind speed, precipitation, surface air pressure, low-level wind speed, cloud cover, and snow cover. Photosynthetically active radiation (PAR) in MEGANv3.2 was obtained by dividing the surface downward shortwave radiation by two. The temporal resolution of these variables is either 1-hourly or 3-hourly, and the 3-hourly data are linearly interpolated to the uniform 1-hourly data. All selected parameters were further interpolated from the original $0.5° \times 0.625°$ to a spatial resolution of $0.5° \times 0.5°$ (consistent with the resolution of the vegetation datasets) for driving the MEGANv3.2 model. In our study, MERRA-2 data from 2001 to 2020 were used.

### 2.2.3 Observations

We used in situ observations of BVOC emission fluxes collected from the literature for the comparison with our model estimates. In total, isoprene emission fluxes at 26 observation sites and monoterpene emission fluxes at 11 observation sites were collected and listed in Table S1 and S2. Other BVOCs emission fluxes were not collected, mainly due to their few observations and small contributions to the total BVOC burdens. The units of all collected data were converted to mg C m$^{-2}$ day$^{-1}$ for easy comparison.

In addition, we also employed global isoprene burden data from space-based observations for the comparison with the simulation. The space-based observations from the Cross-track Infrared Sounder (CrIS) include direct retrievals of global isoprene column burden with an optimal estimate for January, April, July, and October of 2013 (Fu et al., 2019; Wells et al., 2020). Although isoprene column burdens are different from emission fluxes, there is a strong positive correlation between them, and thus isoprene burden data can provide a good reference for qualitative analysis of the spatial distribution of isoprene emissions.

### 2.3 Simulations

To isolate the contribution of different influencing factors (vegetation, meteorology, and $CO_2$) to BVOC emission trends from 2001 to 2020, we perform nine sensitivity experiments (Table 1). These experiments consist of two groups. The first group contains four experiments: EMIT_ALL is the control experiment that considers the historical changes of all factors. EMIT_VEG, EMIT_MET, and EMIT_CO$_2$ consider only the historical changes of vegetation parameters, meteorological factors, and $CO_2$ concentration, respectively, while the other factors are fixed as those in 2001. These three experiments were used to quantify the contributions of vegetation, meteorology, and $CO_2$ concentrations to the BVOC emission trends, respectively. In the second group, five experiments were conducted to isolate the contributions of individual vegetation parameters (i.e., PFT, LAIv) and meteorological factors (i.e., temperature, light, and soil moisture). For vegetation parameters, the experimental setup is the same as EMIT_VEG but with PFT (EMIT_VEG_FIX_PFT) or LAIv (EMIT_VEG_FIX_LAIv) fixed as that in 2001. The difference between EMIT_VEG and EMIT_VEG_FIX_PFT/EMIT_VEG_FIX_LAIv represents the contribution of LAIv and PFT historical changes to the BVOC emission trends. For meteorological factors, the experimental

setup is the same as EMIT_MET but with temperature (EMIT_MET_FIX_T2m), light (EMIT_MET_FIX_RAD), or soil moisture (EMIT_MET_FIX_SM) fixed as that in 2001. The difference between EMIT_MET and EMIT_MET_FIX_T2m, EMIT_MET_FIX_RAD, and EMIT_MET_FIX_SM represent the impact of temperature, light, and soil moisture changes to the BVOC emission trends, respectively. The model horizontal resolution is $0.5° × 0.5°$, the temporal resolution is 1 hour, and the simulation period is 2001-2020. The input variables include the satellite-retrieved vegetation parameters and MERRA-2 reanalysis data as described above.

**Table 1: Description of model experiments driven with vegetation parameters, meteorological parameters, and $CO_2$.**

| Simulations | LAIv | PFT | T2m | RAD | SM | $CO_2$ |
|---|---|---|---|---|---|---|
| EMIT_ALL | 2001-2020 | 2001-2020 | 2001-2020 | 2001-2020 | 2001-2020 | 2001-2020 |
| EMIT_VEG | 2001-2020 | 2001-2020 | 2001 | 2001 | 2001 | 2001 |
| EMIT_MET | 2001 | 2001 | 2001-2020 | 2001-2020 | 2001-2020 | 2001 |
| EMIT_CO₂ | 2001 | 2001 | 2001 | 2001 | 2001 | 2001-2020 |
| EMIT_VEG_FIX_PFT | 2001-2020 | 2001 | 2001 | 2001 | 2001 | 2001 |
| EMIT_VEG_FIX_LAIv | 2001 | 2001-2020 | 2001 | 2001 | 2001 | 2001 |
| EMIT_MET_FIX_T2m | 2001 | 2001 | 2001 | 2001-2020 | 2001-2020 | 2001 |
| EMIT_MET_FIX_RAD | 2001 | 2001 | 2001-2020 | 2001 | 2001-2020 | 2001 |
| EMIT_MET_FIX_SM | 2001 | 2001 | 2001-2020 | 2001-2020 | 2001 | 2001 |

Note: LAIv = leaf area index of vegetation covered surfaces, PFT = plant functional type, T2m = 2 m temperature, RAD = surface solar radiation, SM = soil moisture, $CO_2$ = $CO_2$ concentrations.

## 3 Results

### 3.1 Spatio-temporal distribution of BVOC emissions

### 3.1.1 Spatial distribution of BVOC emissions

The latest version of the MEGANv3.2 model used in this study has not been fully evaluated in previous studies, thus we first compared the simulation results with the in-situ isoprene and monoterpene emission fluxes collected from the literature. For a fair comparison, the simulated isoprene and monoterpene fluxes were interpolated to the sample-specific locations and averaged over the same measurement period.

Fig. 1a shows the locations of the observation sites, which are mainly distributed in North America, Europe, and Asia. Four main vegetation types are included: (1) evergreen broadleaf forest (EBF), (2) deciduous broadleaf forest (DBF), (3) evergreen needleleaf forest (ENF), and (4) grassland (Grass). Observations show that there is a large range of isoprene emission fluxes (10-120 mg C m$^{-2}$ day$^{-1}$) and monoterpene emission fluxes (0.1-15 mg C m$^{-2}$ day$^{-1}$) for different PFTs (Guenther et al., 2012). Isoprene emission flux is generally larger in EBF and DBF than in ENF and Grass. Compared to the observations of isoprene emission fluxes, MEGANv3.2 can simulate the magnitude of isoprene emission fluxes with a correlation coefficient of 0.48 and a mean bias of -3.25 mg C m$^{-2}$ day$^{-1}$ (Fig. 1b). The model can also simulate the larger isoprene emission fluxes for EBF and DBF than for ENF and Grass. However, the model tends to underestimate isoprene fluxes from grassland by a factor of about 10, possibly because the prescribed grass emission factors are too low. The model also tends to mostly overestimate the isoprene emission fluxes from EBF while underestimate them from DBF. For ENF, although there is only one station, the model significantly overestimates the isoprene emission fluxes by a factor of 6.

Fewer monoterpene samples (11) were collected than isoprene samples (26). As shown in Fig. 1c, the model overestimates the monoterpene emission with a correlation coefficient of 0.34 and a mean bias of 3.65 mg C m$^{-2}$ day$^{-1}$. Although the overestimation is mostly within a factor of 10, it largely overestimates the monoterpene emission flux at two sites located in Eastern China and Northwestern South America by two orders of magnitude. Comparison of long-term observations at the K34 tower site (vegetation type: EBF) in 2013 in the Amazon reveals that the simulated seasonal variations of isoprene emission fluxes are similar to the observation (Fig. 1d). The model also captures the increase in emission during the dry season with a correlation coefficient of 0.48, although there is a smaller contrast between dry season and other seasons in the model.

Not surprisingly, there are still large discrepancies between the simulations and observations, which may be ascribed to the deficiency in emission parameterizations such as vegetation emission factors. It may be also ascribed to input parameters such as vegetation and meteorology. In addition, note that the comparison of in situ isoprene measurements with the model is not always representative. Isoprene has a very short lifetime (minutes to hours) in the atmosphere, which implies that its measured fluxes depend on the local tree or plant species near the observation site. The model results at a horizontal resolution of 50 km represent a regional mean for the grid cell where the specific observation site is located, which can also partly explain the difference between the simulations and observations because of the large spatial variability in BVOC emissions. The large bias in the seasonal variation of isoprene fluxes in MEGANv3.2 may be due to a lack of representation of the isoprene emission capacity of tree species at different leaf ages (Alves et al., 2018). Additionally, the model bias arises from a lack of realistic representations of leaf phenology, canopy structure, soil moisture feedbacks, and variation in isoprene emissions due to the complex biodiversity in the Amazon region.

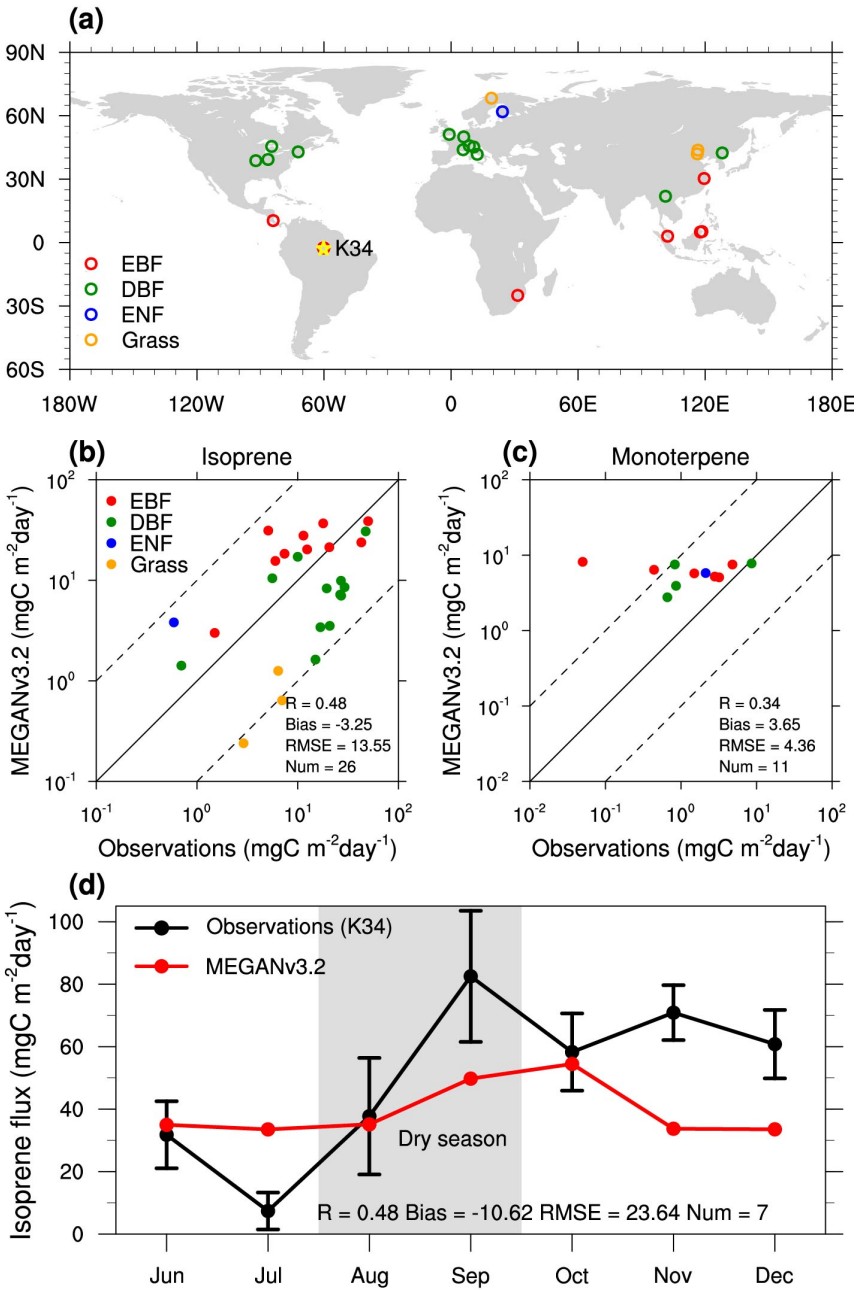

**Figure 1: (a)** The location distribution of isoprene and monoterpene observation sites based on the literature collection (the five-pointed star represents the location of the K34 tower site, EBF: evergreen broadleaf forest, DBF: deciduous broadleaf forest, ENF: evergreen needleleaf forest, Grass: grassland), and comparison with simulated **(b)** isoprene fluxes, **(c)** monoterpene fluxes, and **(d)** monthly variations of isoprene fluxes measured at the K34 tower site in 2013 (the upper and lower limits of the observations represent one standard deviation). R: correlation coefficient, Bias: absolute bias, RMSE: Root mean square error, and Num: Total number of observation samples. In (b) and (c), the 1:1 (solid) and 1:10/10:1 (dashed) lines are plotted for references. In (d), gray shade denotes the dry season.

Fig. 2 shows the comparison of simulated isoprene emission fluxes from this study, IASB-TD-OMI dataset, and CrIS. IASB-TD-OMI datasets employ a "top-down" approach to constrain isoprene emission fluxes simulated by a chemical transport model with formaldehyde observations from the Ozone Monitoring Instrument (OMI) (Stavrakou et al., 2014, 2015). CrIS is directly retrieved from satellite observations (Section 2.2). Note that CrIS provides the isoprene burden, which cannot be directly compared with the modeled emission flux. Here we use it to indicate the relative intensity of isoprene emission (Section 2.2) and validate the spatial patterns of simulated isoprene emissions.

The simulated results and IASB-TD-OMI show similar spatial patterns of isoprene emission, with the correlation coefficients of 0.64, 0.77, 0.74 and 0.77 for January, April, July, and October, respectively. However, except for January, the simulated emissions from this study are systematically higher than those obtained from IASB-TD-OMI by about 20%. The differences are mainly concentrated in South America, Central Africa, and Southeast Asia, which may be partly due to differences in estimation methods (i.e., top-down vs. bottom-up), emission model parameters (e.g., vegetation emission factors) and the meteorological datasets used (MERRA-2 vs. ERA-Interim). In addition, the simulated isoprene emissions are compared to the isoprene column burdens retrieved from the CrIS. The spatial distributions of simulated isoprene emission and the CrIS-retrieved column burden are moderately correlated, with the correlation coefficient varying between 0.43 and 0.58. The isoprene emission and the column burden show similar spatial patterns (e.g., large values in Eastern North America and Central Africa) and seasonal variations (e.g., the lowest emission flux and column burden in July), which suggests that the model estimates reasonable isoprene emissions.

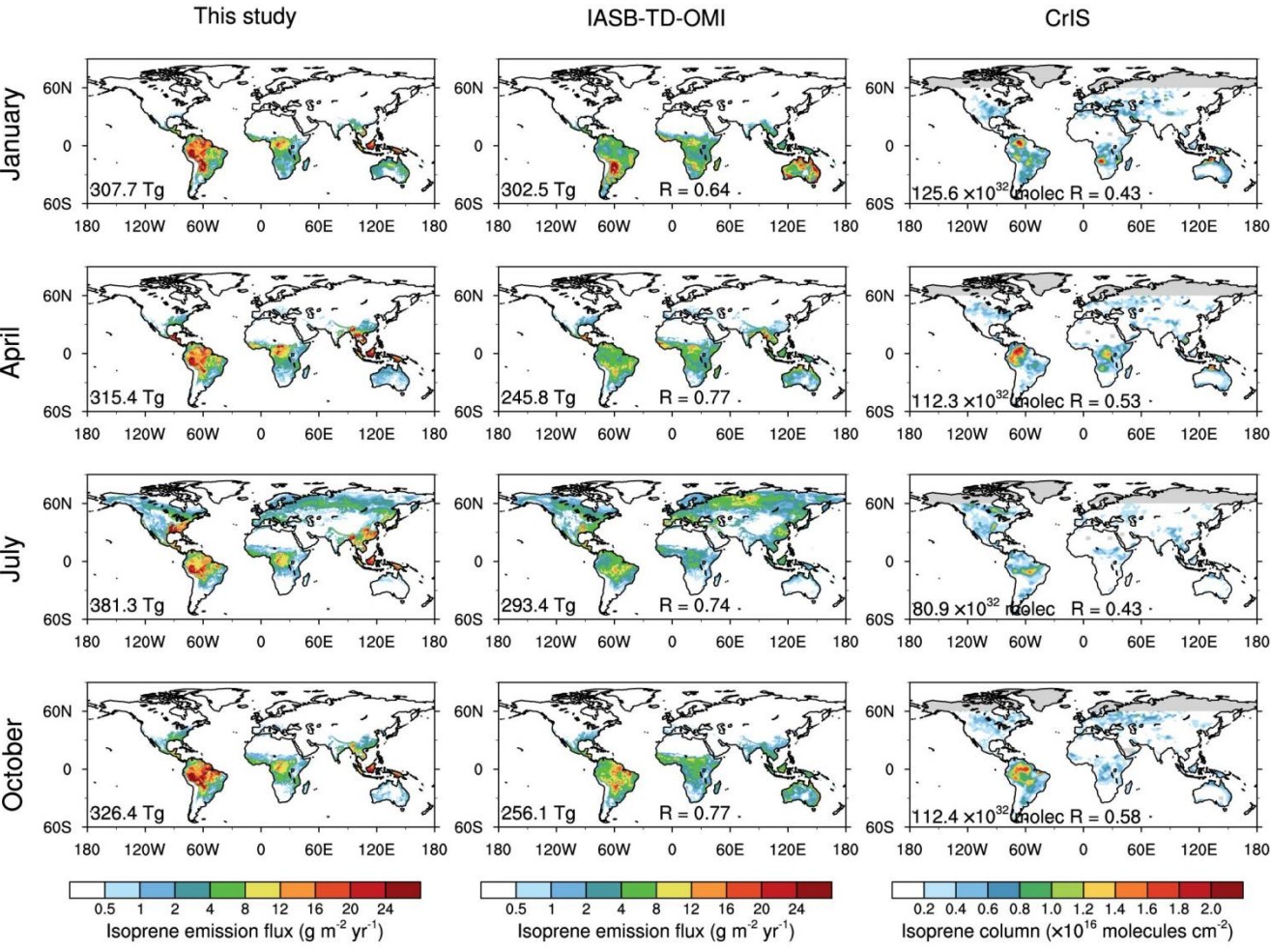

**Figure 2: The comparison of (left column) simulated isoprene emission fluxes (0.5° × 0.5°) with (middle column) IASB-TD-OMI (0.5° × 0.5°) and (right column) CrIS data (2° × 2.5°) in January, April, July, and October of 2013. The annual global emission or burden are given at the bottom-left of each panel. The correlation coefficients R of the simulation results with IASB-TD-OMI and CrIS are marked in the subplots in the middle and right columns, respectively. Gray shade denotes the region where data is not available.**

Fig. 3 shows the spatial distribution of annual emission fluxes for four BVOC categories, including isoprene, monoterpenes, sesquiterpenes, and other BVOCs. The contribution of each category to total BVOC emission are also shown. The spatial patterns of the emission fluxes for the four BVOC categories are relatively similar (Fig. 3a-d). The strongest emissions (>20 g m$^{-2}$ yr$^{-1}$ for total BVOC emission) are mainly located in the tropical regions such as South America, Central Africa, and Southeast Asia. The latitudinal distribution of emissions shows that the peaks of all four categories are located at 0-15°S, decreasing gradually toward the poles (Fig. 3f). Isoprene emission accounts for the largest fraction (> 40%) of total BVOC emission at 0-15°S and the fraction decreases in higher latitudes. In contrast, monoterpene emission accounts for the

least fraction (around 20%) at 0-15 °S and the fraction increases to ~40% in high altitudes. The latitudinal variations of different BVOC categories are mainly caused by different PFT covers at different latitudes.

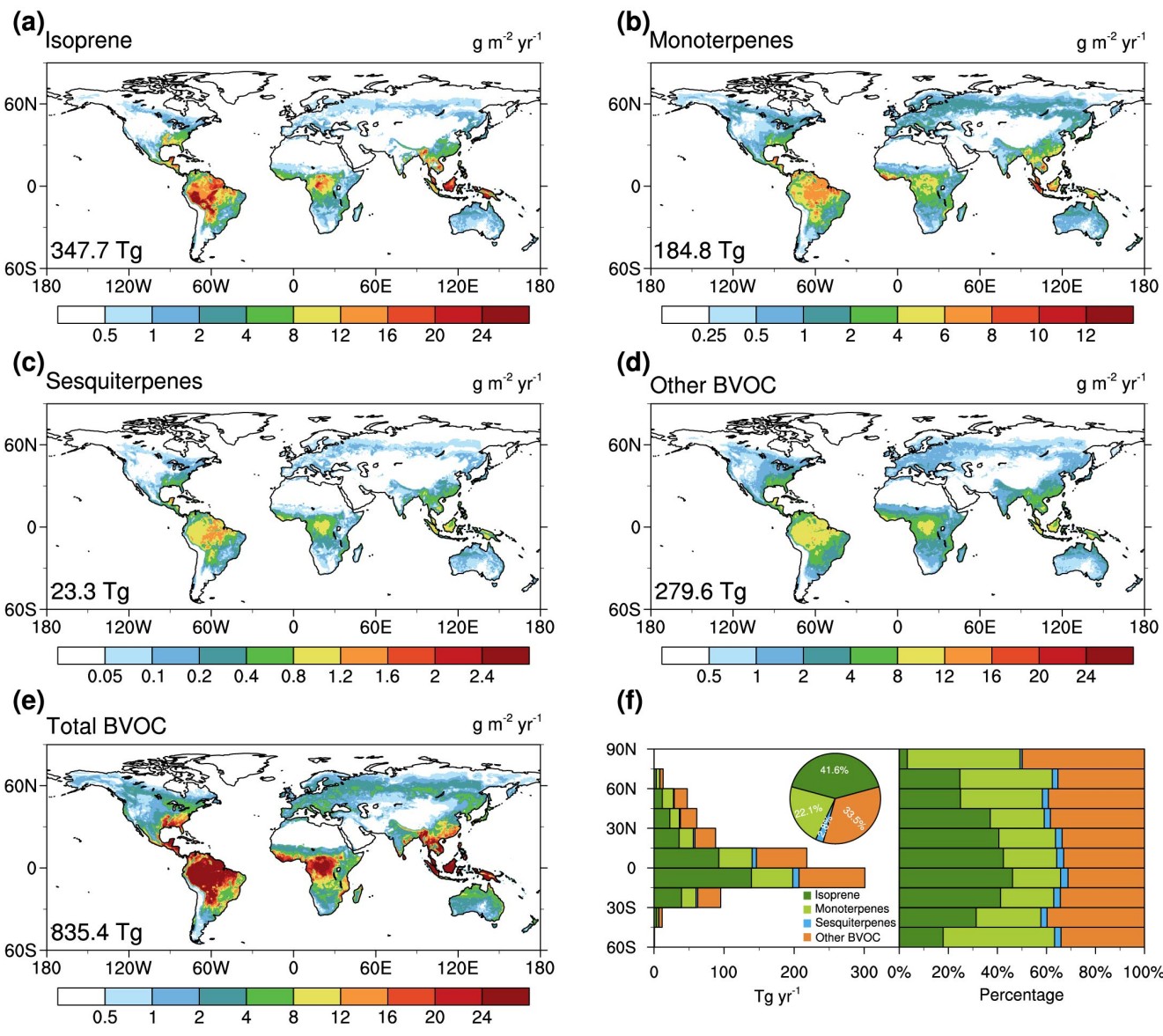

**Figure 3: The spatial pattern of the annual average BVOC emission fluxes of (a) isoprene, (b) monoterpenes, (c) sesquiterpenes, (d) other BVOCs, (e) total BVOC, and (f) the latitudinal distribution (left: total emission, right: fraction of each category; pie chart: the fraction of each category in global total emissions) for the period 2001–2020. The annual global emissions are given at the bottom-left of each subplot.**

In total, annual global BVOC emission is 835.4 Tg yr$^{-1}$ during 2001-2020. The total emission for isoprene, monoterpenes, sesquiterpenes, and other BVOC is 347.7 Tg yr$^{-1}$, 184.8 Tg yr$^{-1}$, 23.3 Tg yr$^{-1}$, and 279.6 Tg yr$^{-1}$, respectively, and accounts for 41.6%, 22.1%, 2.8%, and 33.5%, respectively of the total BVOC emission. Overall, isoprene is the dominant component for total BVOC emissions. The contribution from sesquiterpenes is relatively minor compared to isoprene and monoterpenes emissions. Table 2 further compares the BVOC emissions estimated in this study with others. The annual global BVOC emission (835.4 Tg yr$^{-1}$) from this study is within the range (558-1005 Tg yr$^{-1}$) of previous estimations with MEGANv2.1 using different drivers. The annual global isoprene emission estimated in this study (347.7 Tg yr$^{-1}$) is smaller than most previous bottom-up estimates, but is close to the top-down estimate based on GOME2 (344.7 Tg yr$^{-1}$). Interestingly, the isoprene emission estimated in this study is 12% lower than that estimated by Weng et al. (2020), while the monoterpene emission estimated in this study is 26% higher, although the present study used the same driving fields (i.e., MERRA-2 meteorological field and MODIS vegetation parameters) as Weng et al. (2020). These discrepancies are mainly ascribed to the differences in vegetation emission factors between the two versions of MEGAN. As compared to MEGANv2.1, isoprene emission factors are smaller and monoterpene emission factors are larger in MEGANv3.2. Note that MEGANv2.1 only utilizes fixed emission factors corresponding to the PFTs, but the PFT is insufficient to characterize the emission factors, e.g., tree species with the same PFT may have very different BVOC emission rates. MEGANv3.2 further considers differences in emission factors for tree species with the same PFTs. Thus, the vegetation emission factors in MEGANv3.2 are more accurately represented. However, we note that the uncertainties associated with emission factors are still large due to the limited observational data (Guenther et al., 2020). The estimated sesquiterpenes emission in this study (23.3 Tg yr$^{-1}$) is consistent with previous estimates based on the MERRA meteorological field (20.0-21.6 Tg yr$^{-1}$), but is higher than those based on the ERA meteorological field (11.9-16.6 Tg yr$^{-1}$). This is partly due to the higher values of 2 m temperature and downward shortwave radiation field in MERRA than in ERA in the tropical regions, resulting in higher emission (Sindelarova et. al., 2022). In addition, the other BVOC emission estimated in this study (279.6 Tg yr$^{-1}$) is higher than previous studies (115.5-278.8 Tg yr$^{-1}$), which can be partly attributed to more BVOC components (e.g., butane, butanenitrile, acetophenone, benzene cinnamaldehyde, cinnamic acid, etc.) considered in MEGANv3.2 than in MEGANv2.1.

**Table 2. Comparison of annual global BVOC emission rate with previous studies (Tg yr$^{-1}$).**

| Reference | Period | Method (model, meteorology, vegetation) | Isoprene | Monoterpene | Sesquiterpenes | other BVOC | Total BVOC |
|---|---|---|---|---|---|---|---|
| **Bottom-up** | | | | | | | |
| **This study** | **2001-2020** | **MEGANv3.2, MERRA-2, MODIS** | **347.7** | **184.8** | **23.3** | **279.6** | **835.4** |
| Sindelarova et al. (2022) | 2000-2019 | MEGANv2.1, ERA5, CLM4 | 440.5 | 82.7 | 16.6 | 222.3 | 762.1 |
| Sindelarova et al. (2022) | 2000-2019 | MEGANv2.1, ERA5, ESA-CCI | 299.1 | 63.2 | 11.9 | 183.7 | 557.9 |
| Sindelarova et al. (2022) | 2000-2017 | MEGANv2.1, ERA-Interim, CLM4 | 385.2 | 78.5 | 14.9 | 211.4 | 690.0 |

| Reference | Period | Model/Data | | | | | |
|---|---|---|---|---|---|---|---|
| Weng et al. (2020) | 1980–2017 | MEGANv2.1 in HEMCO, MERRA-2, MODIS | 391.0 | 135.9 | 21.6 | 115.5 | 664.0 |
| Weng et al. (2020) | 2014-2017 | MEGANv2.1 in HEMCO, GEOS-FP, MODIS, 4°× 5° | 374.0 | 142.8 | 23.3 | 124.9 | 665.0 |
| Weng et al. (2020) | 2014-2017 | MEGANv2.1 in HEMCO, GEOS-FP, MODIS, 2°× 2.5° | 377.4 | 140.4 | 22.7 | 121.3 | 661.8 |
| Weng et al. (2020) | 2014-2017 | MEGANv2.1 in HEMCO, GEOS-FP, MODIS, 0.25°× 0.3125° | 386.5 | 135.8 | 21.8 | 115.9 | 659.9 |
| Sindelarova et al. (2014) | 1980–2010 | MEGAN-MACC, MERRA, MODIS | 594.0 | 95.0 | 20.0 | 261.3 | 970.3 |
| Guenther et al. (2012) | 2000 | MEGANv2.1, Qian 2006, CLM4-SP | 535.0 | 162.3 | 29.0 | 278.8 | 1005.1 |
| Opacka et al. (2021) | 2001-2016 | MEGAN-MOHYCAN, ERA-Interim, CLM4 | 418.0 | | | | |
| Opacka et al. (2021) | 2001-2016 | MEGAN-MOHYCAN, ERA-Interim, MODIS | 520.0 | | | | |
| Opacka et al. (2021) | 2001-2016 | MEGAN-MOHYCAN, ERA-Interim, GFW and MODIS | 354.0 | | | | |
| Arneth et al. (2011) | 1981–2002 | LPJ-GUESS, CRU, LPJV | 524.7 | | | | |
| Arneth et al. (2011) | 1981–2002 | MEGANv2.02, NCEP, MODIS | 428.4 | | | | |
| Arneth et al. (2011) | 1981–2002 | BVOCEM, UM, SDGVMV | 533.8 | | | | |
| Guenther et al. (2006) | 2003 | MEGANv2.02, NCEP, MODIS | 600.0 | | | | |
| **Top-down** | | | | | | | |
| Stavrakou et al., (2015) | 2005-2014 | OMI-based, MEGAN-MOHYCAN, ERA-Interim, MODIS | 273.9 | | | | |
| Stavrakou et al., (2014) | 2007-2012 | GOME2-based, MEGAN-MOHYCAN, ERA-Interim, MODIS | 344.7 | | | | |
| Shim et al., (2005) | 1996-1997 | GOME-based, GEOS-Chem-MEGAN, GEOS-STRAT, AVHRR | 641.5 | | | | |

Note: MEGAN (Model of Emission of Gases and Aerosols from Nature), HEMCO (Harvard-NASA Emissions Component), MACC (Monitoring Atmospheric Composition and Climate project), MOHYCAN (Model of Hydrocarbon emissions by the CANopy), LPJ-GUESS (Lund-Potsdam-Jenna General Ecosystem Simulator), BVOCEM (Biogenic Volatile Organic Compound Emission Model), OMI (Ozone Monitoring Instrument), GOME (Global Ozone Monitoring Experiment instrument), GEOS-Chem (the global 3-D model of atmospheric chemistry driven by meteorological input from the Goddard Earth Observing System (GEOS) of the NASA Global Modeling and

Assimilation Office (GMAO)), MERRA (the Modern-Era Retrospective analysis for Research and Applications), ERA5 (the European Center for Medium-Range Weather Forecasting fifth generation of atmospheric reanalysis products), ERA-Interim (the European Center for Medium-Range Weather Forecasting interim reanalysis products), GEOS-FP (GEOS-Chem met field archive of the GMAO "forward processing" product), Qian 2006 (Qian et al. (2006) atmospheric forcing), CRU (the Climatic Research Unit of the University of East Anglia), NCEP (the National Center for Environmental Prediction reanalysis product), UM (climate model output from the UK Met Office Unified

Model), GEOS-STRAT (met field data product compatible with GEOS-Chem from GMAO), MODIS (Moderate-resolution Imaging Spectroradiometer), CLM4 (Community Land model), ESA-CCI (Climate Change Initiative of the European Space Agency) CLM-SP (standard global simulation constrained by observed land cover), GFW (Global Forest Watch), LPJV (Lund-Potsdam-Jenna vegetation), SDGVMV (Sheffield Dynamic Vegetation Model), AVHRR (Advanced Very High Resolution Radiometer).

### 3.1.2 Seasonal variation of isoprene emissions

The seasonal cycle of simulated isoprene emissions over the nine regions (i.e., NAM: North America, EUR: Europe, NAS: North Asia, EAS: East Asia, SAS: South Asia, SEAS: Southeast Asia, SAM: South America, CAF: Central Africa, AUS: Australia) are presented in Fig. 4. Overall, different regions show different seasonal cycles. Isoprene emissions mainly depend on vegetation and meteorological conditions. At mid to high latitudes, due to the densest vegetation and highest temperatures in the summer months (Fig. S1), isoprene emissions peak in the summer months. In contrast, isoprene emissions are lowest

during winter months in these regions. For example, in North America, Europe, North Asia, and East Asia of the Northern Hemisphere, isoprene emissions are highest in July and lowest in January. We note that the seasonal variations of isoprene emissions in Europe are consistent with the results from the EMEP (European Monitoring and Evaluation Programme) model, which are derived based on the cover fractions and emission factors of detailed tree species and other vegetation (categorized into six PFTs) (Sindelarova et al., 2022). In Australia of the Southern Hemisphere, isoprene emissions peak in austral summer.

At low latitudes of Northern China, South Asia and South America isoprene emissions peak at the end of the dry season in May and October, respectively due to the higher temperatures (Fig. S1). In the tropical regions and low latitudes of the Southern Hemisphere, including Southeast Asia and Central Africa, isoprene emissions exhibit a typical bimodal distribution with two peaks in April and October, corresponding to the peak LAI and temperature in these two months (Fig. S1).

      In total, the global total isoprene emission peaks in July and reaches its minimum in January. This is different from the

study of Sindelarova et al. (2014), which showed the peak of global total isoprene emission in October and the lowest value in June, mainly because they simulated greater isoprene emissions over South America and Central Africa than our study.

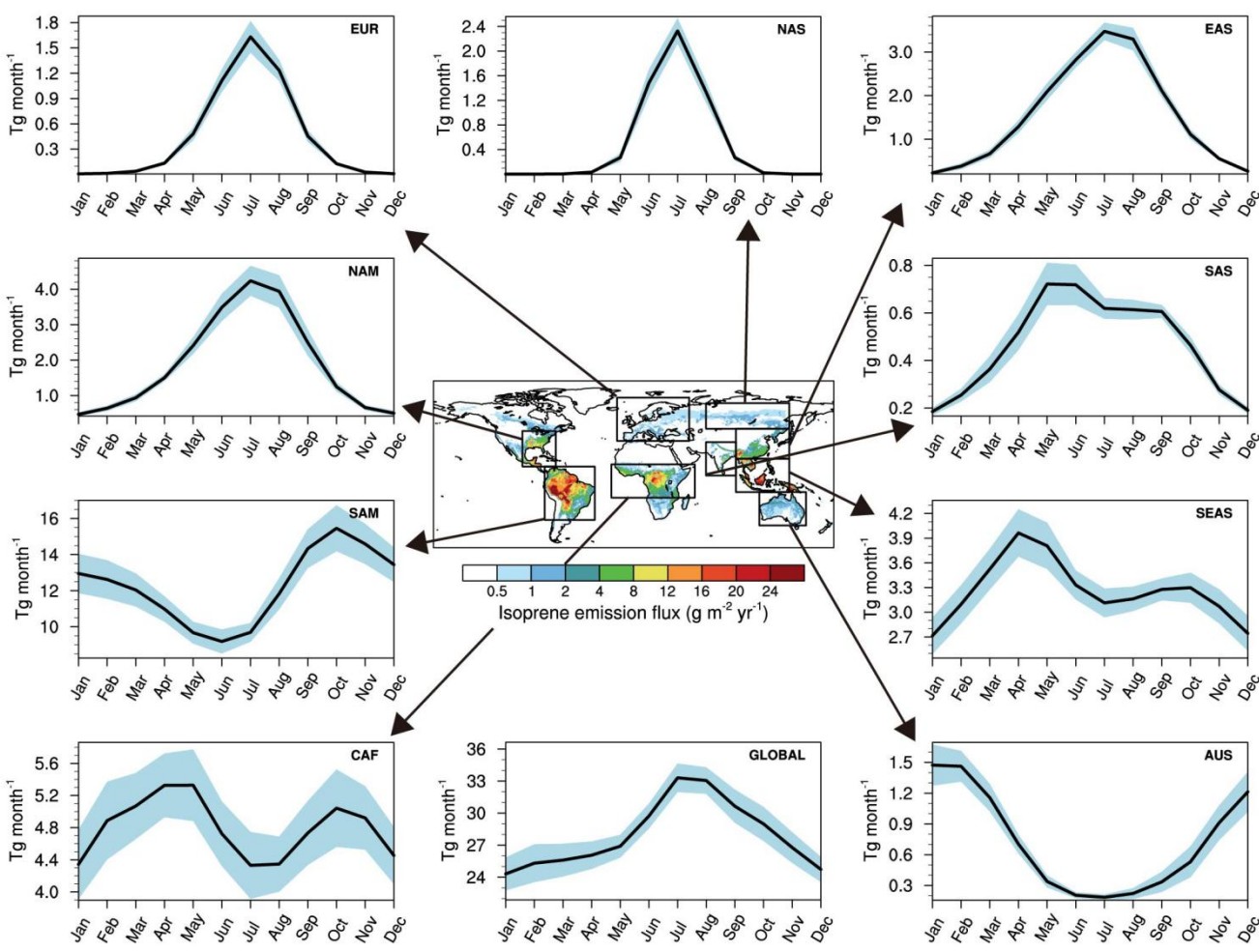

**Figure 4: The seasonal variation of isoprene emissions for each region outlined in black on the map averaged during 2001-2020. The shaded area represents one standard deviation. The nine regions are listed below: NAM: North America; EUR: Europe; NAS: North Asia; EAS: East Asia; SAS: South Asia; SEAS: Southeast Asia; SAM: South America; CAF: Central Africa; AUS: Australia.**

### 3.2 Trends of BVOC emissions

Fig. 5 shows the interannual variation of total isoprene emissions over the nine regions for the last 20 years from the four experiments in the first group (ALL, VEG, MET, and CO₂). By comparing these experiments, the individual impact of three major factors (i.e., vegetation, meteorology, and $CO_2$) on isoprene emissions can be identified. There are strong interannual variations in isoprene emissions in all the regions. As shown in Table 3, the standard deviation of the regionally-summed isoprene emissions is 0.1-8.1 Tg yr$^{-1}$ in the nine regions, and the ratio of standard deviation to climatology mean is 0.01-0.08. For the trends, although the trend in global isoprene emissions between 2001 and 2020 is weak (~0.07% yr$^{-1}$; p=0.67), significant trends in regional emissions can be found in some specific regions. In Europe, East Asia, and South Asia, regionally-

summed isoprene emissions exhibit significantly increasing trends of about $0.37 - 0.66\%$ $yr^{-1}$. In contrast, there is a

significantly decreasing trend in Central Africa ($-0.74\%$ $yr^{-1}$). The trends are weak in the other regions (North America, South America, Southeast Asia, and Australia) except in North Asia, where a decreasing trend of $-0.32\%$ $yr^{-1}$ is modest, but not statistically significant due to the large interannual variability of isoprene emission in this region.

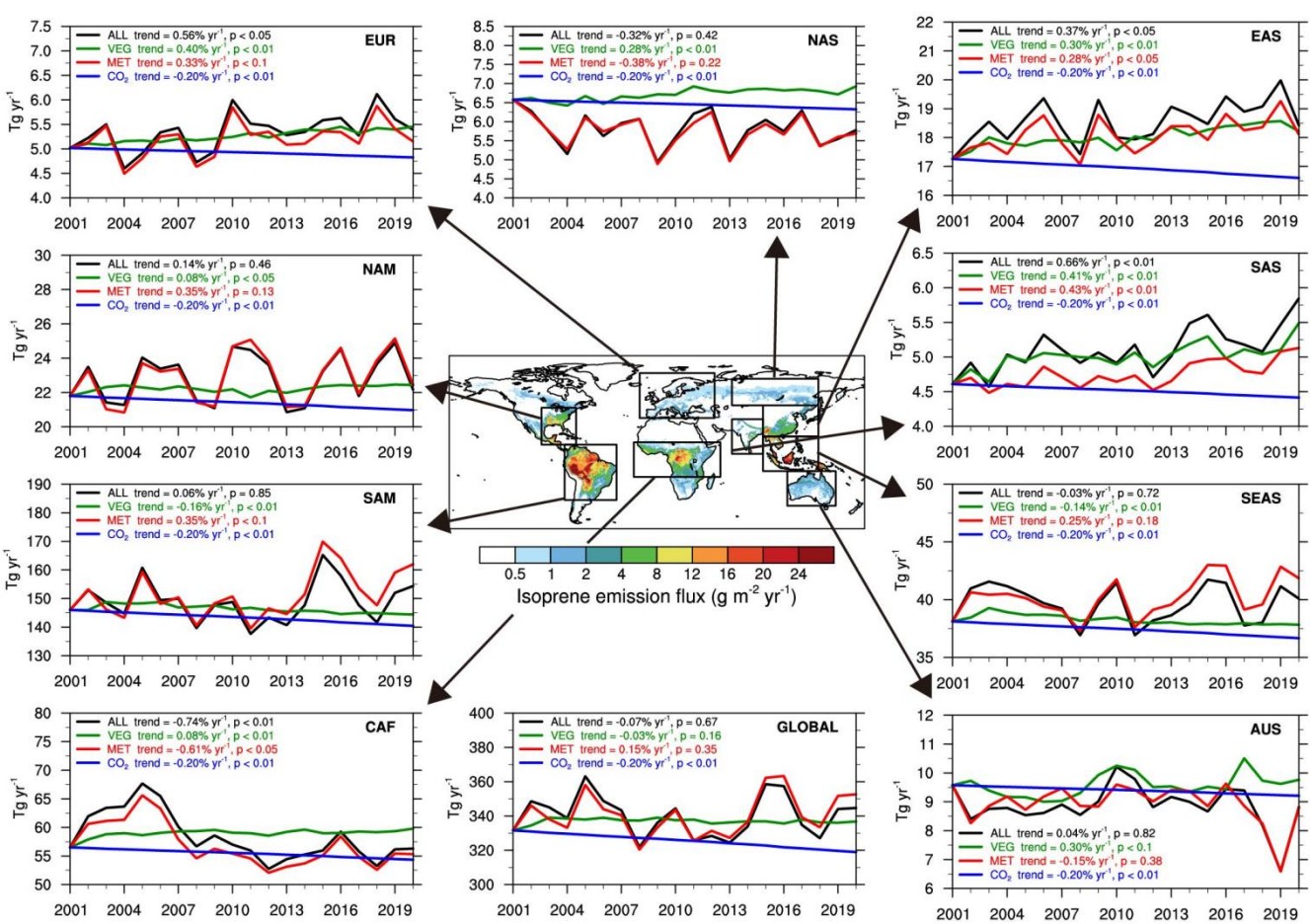

Figure 5: Interannual variation and trends in regional isoprene emissions for each region outlined by black rectangles on the map. Trends are expressed by the relative change in percentage (i.e., the linear change during 2001-2020 divided by the mean value). p-values denote statistically significant levels using the Mann-Kendall test. ALL represents simulated results considering interannual variability of all drivers (i.e., vegetation, meteorology, and $CO_2$), while VEG, MET, and $CO_2$ represent simulated results considering only interannual variability of vegetation, meteorology, and $CO_2$ concentrations, respectively.

**Table 3. The statistical parameters (Avg: average value, Tg yr$^{-1}$; Std: standard deviation, Tg yr$^{-1}$; Std/Avg: ratio of standard deviation to average value) of total isoprene emissions over the nine regions during 2001-2020 from the four experiments in the first group (ALL, VEG, MET, and CO$_2$).**

| NAM | ALL | VEG | MET | CO$_2$ | EUR | ALL | VEG | MET | CO$_2$ | NAS | ALL | VEG | MET | CO$_2$ |
|---|---|---|---|---|---|---|---|---|---|---|---|---|---|---|
| Avg | 22.8 | 22.2 | 22.9 | 21.4 | Avg | 5.3 | 5.3 | 5.2 | 4.9 | Avg | 5.8 | 6.7 | 5.8 | 6.5 |
| Std | 1.4 | 0.2 | 1.5 | 0.3 | Std | 0.4 | 0.1 | 0.3 | 0.1 | Std | 0.5 | 0.2 | 0.4 | 0.1 |
| Std/Avg | 0.06 | 0.01 | 0.06 | 0.01 | Std/Avg | 0.07 | 0.02 | 0.07 | 0.01 | Std/Avg | 0.08 | 0.02 | 0.08 | 0.01 |

| EAS | ALL | VEG | MET | CO$_2$ | SAS | ALL | VEG | MET | CO$_2$ | SEAS | ALL | VEG | MET | CO$_2$ |
|---|---|---|---|---|---|---|---|---|---|---|---|---|---|---|
| Avg | 18.6 | 18.0 | 18.1 | 16.9 | Avg | 5.6 | 5.5 | 5.3 | 5.0 | Avg | 39.7 | 38.3 | 40.2 | 37.4 |
| Std | 0.7 | 0.4 | 0.6 | 0.2 | Std | 0.3 | 0.2 | 0.2 | 0.1 | Std | 1.6 | 0.4 | 1.7 | 0.5 |
| Std/Avg | 0.04 | 0.02 | 0.03 | 0.01 | Std/Avg | 0.06 | 0.04 | 0.04 | 0.01 | Std/Avg | 0.04 | 0.01 | 0.04 | 0.01 |

| SAM | ALL | VEG | MET | CO$_2$ | CAF | ALL | VEG | MET | CO$_2$ | AUS | ALL | VEG | MET | CO$_2$ |
|---|---|---|---|---|---|---|---|---|---|---|---|---|---|---|
| Avg | 148.8 | 146.3 | 151.2 | 143.3 | Avg | 58.3 | 59.0 | 56.9 | 55.5 | Avg | 8.9 | 9.6 | 8.9 | 9.4 |
| Std | 7.0 | 1.4 | 8.1 | 1.7 | Std | 4.1 | 0.7 | 3.7 | 0.7 | Std | 0.7 | 0.4 | 0.7 | 0.1 |
| Std/Avg | 0.05 | 0.01 | 0.05 | 0.01 | Std/Avg | 0.07 | 0.01 | 0.07 | 0.01 | Std/Avg | 0.08 | 0.04 | 0.08 | 0.01 |

For the impacts of various factors, the interannual variations of isoprene emissions are mainly determined by meteorological factors in all the regions, as the time series of isoprene emissions in MET follows closely with those in ALL. In comparison, vegetation factors play a much smaller role in the interannual variations of isoprene emissions. As CO$_2$ concentrations increase constantly, there is little impact from CO$_2$ concentrations on the interannual variations of isoprene emissions. The most remarkable impacts of vegetation and CO$_2$ concentrations is on the trends of isoprene emissions during
the last 20 years.

    Vegetation factors lead to significant trends of isoprene emissions ($p<0.1$ for AUS; $p<0.05$ for other regions) in all regions. In Europe, North Asia, East Asia, South Asia, and Australia, vegetation changes (increases in vegetation cover or shift in vegetation type) strongly promote isoprene emissions, with the trends of $0.28\% - 0.41\%$ yr$^{-1}$. In contrast, vegetation changes (decreases in vegetation cover or shift in vegetation type) reduce isoprene emissions in South America and Southeast
Asia (with the trends of about $-0.15\%$ yr$^{-1}$), which may be ascribed to the local deforestation.

    There are also considerable contributions from meteorological factors on the trends of isoprene emissions. Changes in meteorological factors lead to significantly increasing trends of isoprene emissions ($0.28\%$ to $0.43\%$ yr$^{-1}$; $p<0.1$) in Europe, East Asia, South Asia, and South America. In contrast, meteorological factors cause a strongly declining trend of isoprene emissions in Central Africa ($-0.61\%$ yr$^{-1}$; $p<0.05$). There are also increasing trends of isoprene emissions ($0.25\%$ to

0.35% $yr^{-1}$) in North America and Southeast Asia and decreasing trends ($-0.15\%$ to $-0.38\%$ $yr^{-1}$) in North Asia and Australia due to changes in meteorological factors, although the trends are not statistically significant ($p>0.1$).

As expected, the $CO_2$ inhibition effect results in a significant decrease of $-0.20\%$ $yr^{-1}$ in isoprene emissions (Fig. 5). Note that this study uses a globally uniform and yearly mean $CO_2$ concentration without considering spatial and seasonal variations of $CO_2$ concentration. Additionally, the $CO_2$ concentration can also indirectly affect isoprene emissions by changing 380    meteorological and vegetation factors. Specifically, the increase of $CO_2$ concentration is partly responsible for global warming and thus higher temperatures (e.g. Monson et al., 2007), and the increased $CO_2$ concentration can potentially lead to a larger LAIv (Monson et al., 2007). These indirect effects are not explicitly considered in this study.

Different from isoprene emissions, there is no statistically significant effect of $CO_2$ concentration on monoterpene emissions as suggested by previous studies (Malik et al., 2019, 2023). Therefore, monoterpene emissions in MEGANv3.2 only 385    consider the effects of vegetation and meteorological factors and show a significantly positive trend of 0.34% $yr^{-1}$ globally (Fig. S2). In total, the increasing trends caused by vegetation changes and meteorological factors are comparable globally. In Europe, East Asia, and South Asia, the increasing trends induced by vegetation changes (increase in vegetation cover or shift in vegetation type) are larger than those induced by meteorology changes, indicating the dominant role of vegetation change in these regions. In North Asia and Australia, vegetation changes lead to increasing trends, while meteorological changes lead 390    to decreasing trends. The overall impacts are the increase of monoterpene emissions, due to the dominant impacts from vegetation changes. In the other regions, changes in meteorological factors dominates over vegetation changes in determining the trends of monoterpene emissions. In particular, in South America and Southeast Asia, changes in meteorological factors lead to strongly increasing trends of monoterpene emissions, which are much larger than those induced by vegetation changes.

Vegetation changes affect isoprene and monoterpene emissions differently, mainly due to differences in the emission 395    capacity (i.e., emission factors, Equation 2) of vegetation types for these two components. Vegetation changes lead to a stronger trend of the monoterpene emission than that of the isoprene emission in some regions. For example, the isoprene-emission trends are 0.08% $yr^{-1}$, 0.40% $yr^{-1}$, 0.28% $yr^{-1}$, 0.30% $yr^{-1}$, and 0.41% $yr^{-1}$, while the monoterpene-emission trends are up to 0.19% $yr^{-1}$, 0.54% $yr^{-1}$, 0.43% $yr^{-1}$, 0.47% $yr^{-1}$, and 0.56% $yr^{-1}$ in North America, Europe, North Asia, East Asia, and South Asia, respectively. However, in Southeast Asia and South America, the effect of vegetation changes on 400    the trends of monoterpene emissions is weaker. In addition, the effect of meteorological changes on monoterpene emissions is found to be weaker than that on isoprene emissions in most regions, especially in Central Africa, where the trend for isoprene emission is $-0.61\%$ $yr^{-1}$, while the trend for monoterpene emission is $-0.26\%$ $yr^{-1}$. Overall, compared to isoprene emissions, a stronger increasing trend and a weaker decreasing trend of monoterpene emissions are found in these hotspot regions when all influencing factors are considered.

## 3.3 Drivers of BVOC emission trends

### 3.3.1 Changes in vegetation factors, meteorology, and $CO_2$ concentrations

BVOC emissions depend on various factors including vegetation parameters, meteorological conditions, and $CO_2$ concentration. Fig. 6 shows the global distribution of the trends in these influencing factors of BVOC emissions from 2001 to 2020, including vegetation parameters (i.e., VCF, LAI), meteorological parameters (i.e., surface 2 m temperature, surface solar radiation, and soil moisture), and $CO_2$ concentrations. Different regions show different VCF trends. A moderate increase of 0.1% to 1% $yr^{-1}$ in VCF can be found in Central North America, East Asia, and India, while a decrease of −0.1% to −1% $yr^{-1}$ exists in Central South America, Central and Southwest Africa, Western Australia, and Central Asia. Decrease of VCF is mainly related to local deforestation or wildfire burning. Specifically, the four main PFTs used in MEGANv3.2 (i.e., tree, shrub, grass, and crop, Equation 2) have shifted significantly in some regions (Fig. 7a-d). Tree covers increase in East Asia and Europe by about $0.3 - 0.5\%$ $yr^{-1}$. Grass covers increase in Central South America, with a maximum increase trend of more than 1% $yr^{-1}$, corresponding to the decrease of tree and shrub covers in these regions. Crop covers also increase in some regions such as Eastern South America and Northern India.

Most vegetated areas are becoming greener (i.e., higher LAI) during 2001-2020, especially in Europe, East Asia, and South Asia, with a positive trend of exceeding 0.02 $m^2$ $m^{-2}$ $yr^{-1}$. The decrease trends of LAI only exist in some isolated regions in Eastern South America, Central Africa, and Western Australia. Note that VCF and LAI reflect two different aspects of vegetation information using different retrieval methods. VCF represents the amount of ground covered by the vertical projection of vegetation, and its variation focuses on the extension or shrinkage of vegetation on the ground surface. While LAI reflects the amount of vegetation biomass, and its variation emphasizes the changes in vegetation biomass content per unit of ground surface area. Therefore, the variations of these two parameters may be different. LAIv in MEGANv3.2 (Equation 3) is calculated by dividing LAI by VCF, representing the leaf area per unit of the canopy area (vegetation covered area), which implicitly reflects the overall growth condition of vegetation (e.g., the number of plant foliage, canopy structure) per unit of vegetation-covered area. Due to the rapid shrinking of vegetation cover, LAIv in South America shows more remarkable increasing trend than LAI does (Fig. 7e).

For meteorological factors, trends in surface 2 m temperature (T2m) vary significantly in different regions. T2m increases with a small trend (0.01 to 0.05 °C $yr^{-1}$) in most regions during 2001-2020. A larger increase of 0.05 to 0.2 °C $yr^{-1}$ occurs in the Arctic and Europe. Despite increasing trends of T2m in most regions, there are also cooling trends of -0.05 to -0.2°C $yr^{-1}$ in some regions such as Northeastern North America, Central Africa, and Central Asia. Large trends in surface solar radiation exist in tropical regions, with significant dimming trends in Central South America and Central Africa, and brightening trends in some parts of Southeast Asia. Soil moisture increases significantly in large parts of Central Africa, South Asia, and Northeast Asia, while it decreases significantly in South America and Australia. Finally, global $CO_2$ levels have increased dramatically

from 370.57 ppm in 2001 to 412.44 ppm in 2020. Overall, these factors have changed significantly over the past 20 years, which has led to significant changes in BVOCs emissions as shown in Section 3.2.

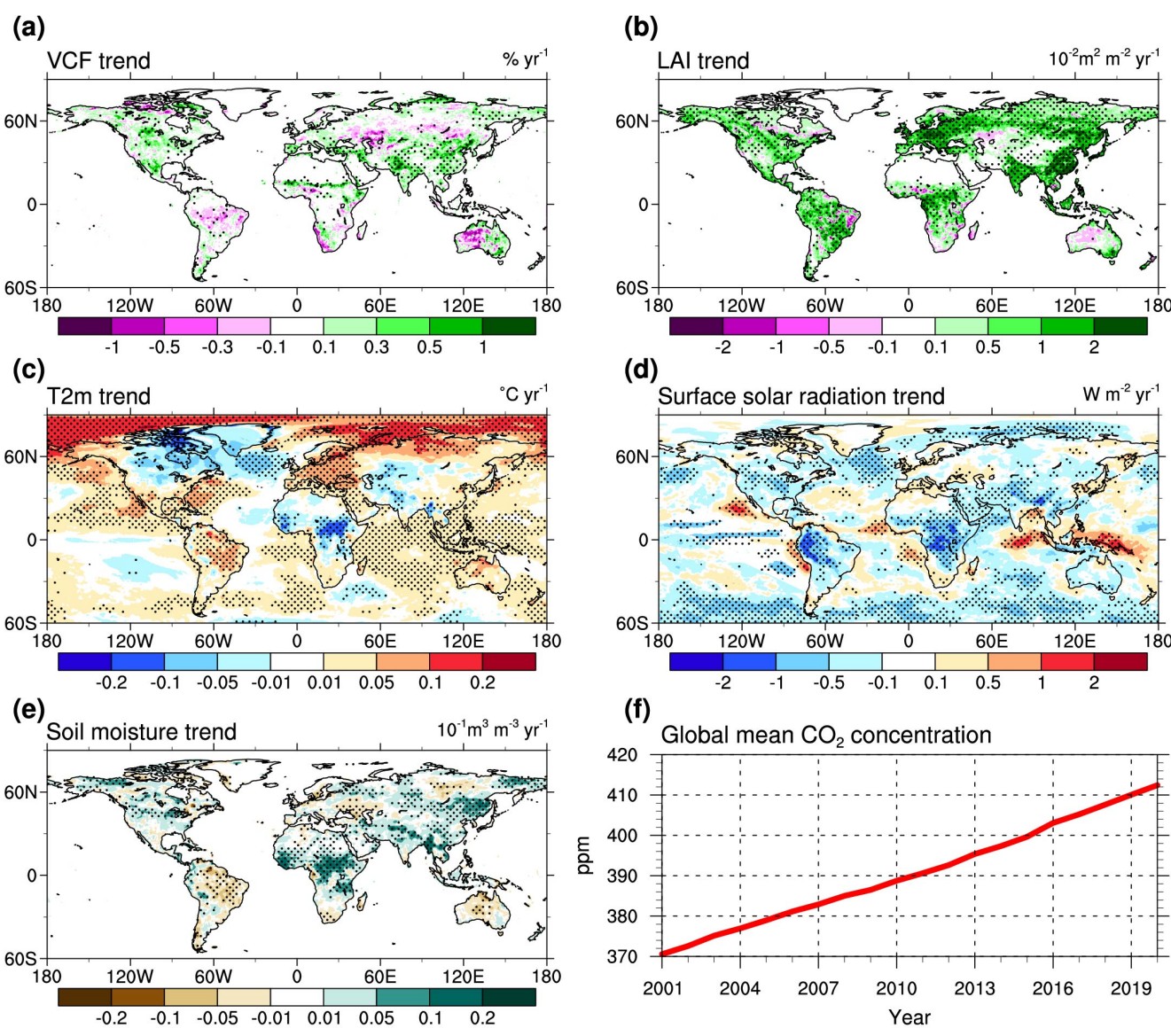

**Figure 6: Trends in (a, b) vegetation factors, (c-e) meteorological factors, and (f) CO₂ concentrations from 2001 to 2020. (a) vegetation cover fraction (VCF), (b) leaf area index (LAI), (c) surface 2 m temperature (T2m), (d) surface solar radiation, (e) soil moisture, and (f) CO₂ concentration. Stippling denotes regions where the trend is statistically significant (p < 0.1) using the Mann-Kendall test.**

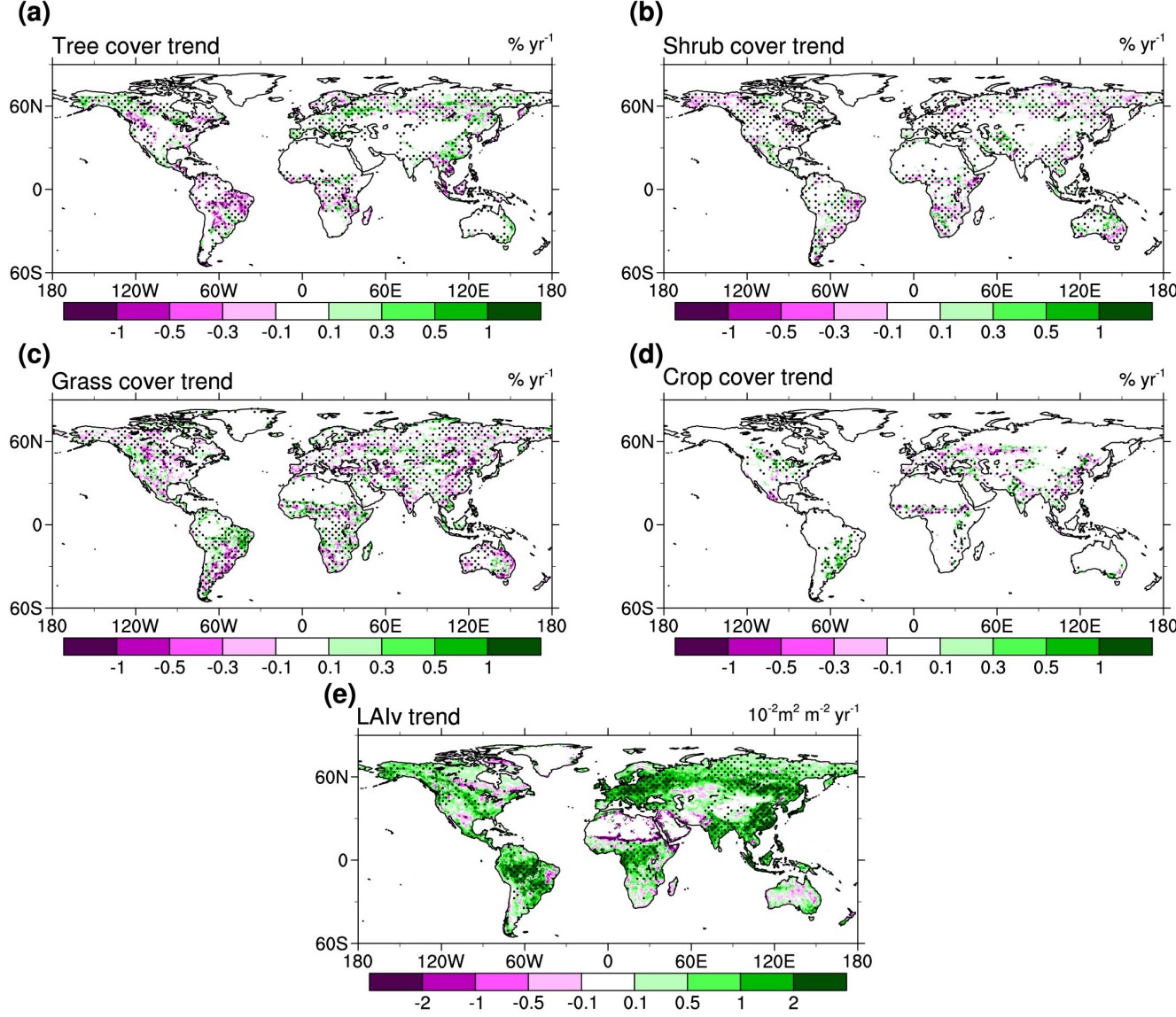

**Figure 7: Trends in (a-d) cover fractions of the four plant functional types (PFTs) and (e) LAIv (leaf area index of vegetation covered surfaces) from 2001 to 2020. (a) tree, (b) shrub, (c) grass, and (d) crop. Stippling denotes regions where the trend is statistically significant (p < 0.1) using the Mann-Kendall test.**

### 3.3.2 Contribution of drivers to BVOC emission trends

We further decompose the main drivers (vegetation, meteorology, and $CO_2$ concentrations) and quantify the contribution of individual driver to isoprene emission trends. Fig. 8 shows the spatial patterns of isoprene emission trends estimated from the nine experiments listed in Table 1. The pattern in Fig. 8a is affected by a combination of vegetation change (Fig. 8b), meteorological variability (Fig. 8c) and $CO_2$ concentration change (Fig. 8d). The statistically significant and positive trends

are mainly located in the mid-latitudes of Eurasia, with the strongest trends of 2% to 5% $yr^{-1}$, while the negative emission trends are distributed in Central Africa and Central Australia, with the largest magnitudes ranging from −5% to −8% $yr^{-1}$. Vegetation changes increase isoprene emissions in most regions except in the Amazon, while $CO_2$ concentration changes decrease isoprene emissions uniformly across the globe (relative trend: −0.20% $yr^{-1}$; absolute trend: −0.7 Tg $yr^{-1}$; p<0.01). For meteorological variability effects, the areas with statistically significant trends are smaller than those considering the effects of vegetation or $CO_2$ concentration changes, and show different signs in different regions. For instance, the positive trends are mainly in Europe, Central Asia, and northern South America, while the negative trends are in Central Africa, Central Australia, and the area around Lake Baikal.

In terms of the effects of individual factors, the trends of isoprene emissions illustrated in Fig. 8e and Fig. 8f are both statistically significant in most areas, which result from the changes in LAIv (Fig. 7e) and PFT (Fig. 7a-d, especially for tree cover). Note that changes in LAIv and PFT lead to opposite trends for the total global isoprene emissions (LAIv effect: 0.07% $yr^{-1}$, p<0.01; PFT effect: −0.1% $yr^{-1}$, p<0.01), resulting in a weak effect of vegetation changes on the global isoprene emission trend (−0.03% $yr^{-1}$; p=0.16). The trends of isoprene emissions illustrated in Fig. 8g-i result from the changes in 2 m temperature (Fig. 6c), surface solar radiation (Fig. 6d), and soil moisture (Fig. 6e), respectively. Temperature changes play a dominant role in the isoprene emission trends, with statistically significant positive trends in Europe and South America, up to 1% to 2% $yr^{-1}$, while negative trends are found in Central Africa, Northwest South Asia, and Southern North Asia, with maximum trends of −2% to −5% $yr^{-1}$. Radiation changes exert a weaker effect on the isoprene emission trends (−0.2% to −0.5% $yr^{-1}$), but show statistically significant negative trends over most regions (global scale, relative trend: −0.06% $yr^{-1}$; absolute trend: −0.2 Tg $yr^{-1}$; p<0.05). In some specific regions, changes of soil moisture significantly affect the trends of isoprene emissions. Positive trends are mainly distributed in Central Africa and Central Asia (2% to 5% $yr^{-1}$), while negative trends are located in Eastern Amazonia and Australia, up to −5% to −10% .

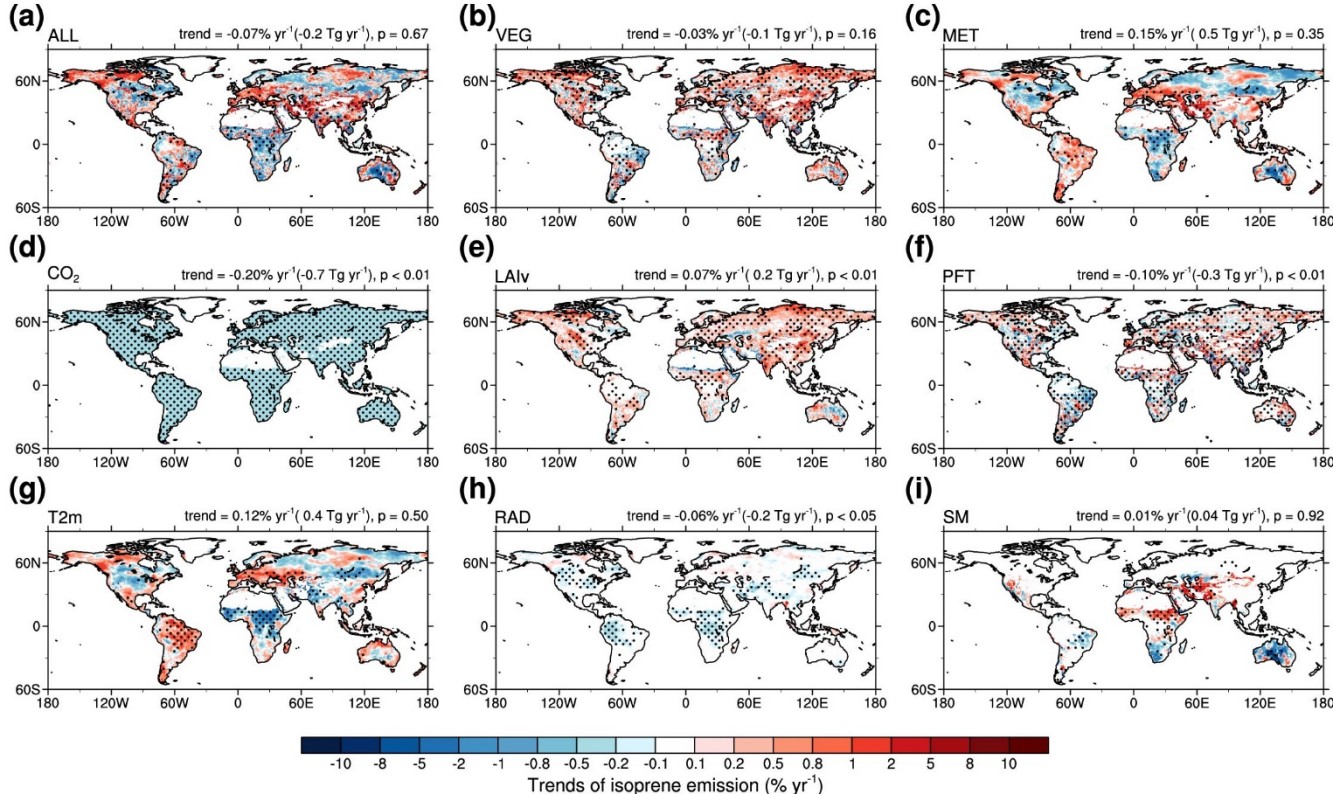

**Figure 8: Spatial distribution of isoprene emission trends from 2001 to 2020. (a) ALL represents simulated results considering interannual variability of all drivers. (b, c, and d) VEG, MET, and CO₂ represent simulated trends when considering only interannual variability of vegetation, meteorology, and CO₂ concentrations, respectively. (e, f) Contributions from individual vegetation parameters including LAIv (leaf area index of vegetation covered surfaces) and PFT (plant functional types). (g, h, and i) Contributions from individual meteorological factors including T2m (surface 2 m temperature), RAD (surface solar radiation), and SM (soil moisture). Stippling denotes regions where the trend is statistically significant (p < 0.1) using the Mann-Kendall test. Trends are expressed by the relative change in percentage (i.e., the linear change during 2001-2020 divided by the mean value) and absolute change. p-values represent statistically significant levels using the Mann-Kendall test.**

To understand the discrepancies in isoprene emission trends across regions and to identify the contribution of individual influencing factors, nine hotspot regions (same as Fig. 4) are selected for further analysis. As shown in Fig. 9a, in some regions of the Northern Hemisphere, such as Europe, East Asia, and South Asia, both vegetation and meteorology changes strongly boost the isoprene emissions. In contrast, isoprene emissions in Central Africa decline sharply, mainly due to the meteorological changes. In other regions, especially Southeast Asia, South America, and Australia, the effects of the three factors offset each other, resulting in overall small trends.

The vegetation parameters can be decomposed into LAIv and PFT cover (Fig. 9b, Equations 2 and 3; the difference between EMIT_VEG and EMIT_VEG_FIX_PFT/EMIT_VEG_FIX_LAIv in Table 1). The LAIv calculated in MEGANv3.2 combines the information from LAI and VCF, which implies the growth of vegetation per unit of vegetation-covered area,

while PFT cover reflects the amount of extension for different vegetation types on the ground surface. Large increases in the isoprene emissions mainly exists in Europe, North Asia, East Asia, South Asia, and Australia, where changes in LAIv and PFT jointly promote the emissions. These regions have experienced a dramatic increase in LAIv, along with an obvious expansion of tree covers. In particular, crop covers in South Asia and shrub covers in Australia increase rapidly (Fig. 9d). In other regions, however, changes in the PFT covers cancel out the contribution from increased LAIv, resulting in weaker positive trends in the isoprene emissions. Southeast Asia and South America even have experienced a moderate decrease of $-0.14\%$ to $-0.16\%$ $yr^{-1}$ in the emissions, mainly due to shifts in local vegetation functional types induced by the deforestation (Fig. 9d). In these regions, primary broadleaf evergreen forests are converted to some economic trees and crops (e.g., rubber, oil palm, and sugar cane).

Three meteorological variables namely temperature, radiation, and soil moisture, which are the main influencing factors from meteorology, are selected for quantifying their effects on the trends of isoprene emissions (Fig. 9c, d; the difference between EMIT_MET and EMIT_MET_FIX_T2m, EMIT_MET_FIX_RAD, and EMIT_MET_FIX_SM in Table 1). The emission trends caused by meteorological factors are large in Europe, East Asia, South Asia, South America, and Central Africa, although the emission trends are not statistically significant in some regions ($p>0.1$). Elevated temperature is found to dominate the rise in emissions in Europe and South America, and temperature cooling dominates the falling in emissions in Central Africa. The enhancement of isoprene emissions in East and South Asia is dominated by the increase of soil moisture. Overall, for meteorological parameters, changes in temperature and soil moisture exert the largest influence on the trends of isoprene emissions, with little effects from changes in radiation except in some regions of South America and Central Africa.

The spatial pattern of monoterpene emission trends is similar to that of isoprene emission trends, but shows stronger positive trends in larger areas, especially in greening hotspots (Fig. S3). The discrepancies are mainly owing to the fact that monoterpene emissions are more sensitive to changes in LAIv (Fig. S3d; LAIv effect: $0.15\%$ $yr^{-1}$, $p<0.01$) and not sensitive to the inhibition effect of $CO_2$. Monoterpene emission trends differ significantly in some regions from isoprene emission trends, which is mainly due to the different impacts of vegetation and meteorological factors on their emissions (Fig. S4). In North America, Southeast Asia, and South America, monoterpene emissions exhibit significantly increasing trends, while isoprene emission trends are not significant. For vegetation parameters, changes in LAIv increase the monoterpene emissions more significantly than on the isoprene emissions, while changes in PFT exert a weaker effect on the monoterpenes emission trends. The combined effects cancel out negative emission trends in Southeast Asia and South America and promote the increase in emissions in other regions. The effect of meteorological factors on monoterpene emission trends is similar to that on the isoprene emission trends, although the effect is weaker.

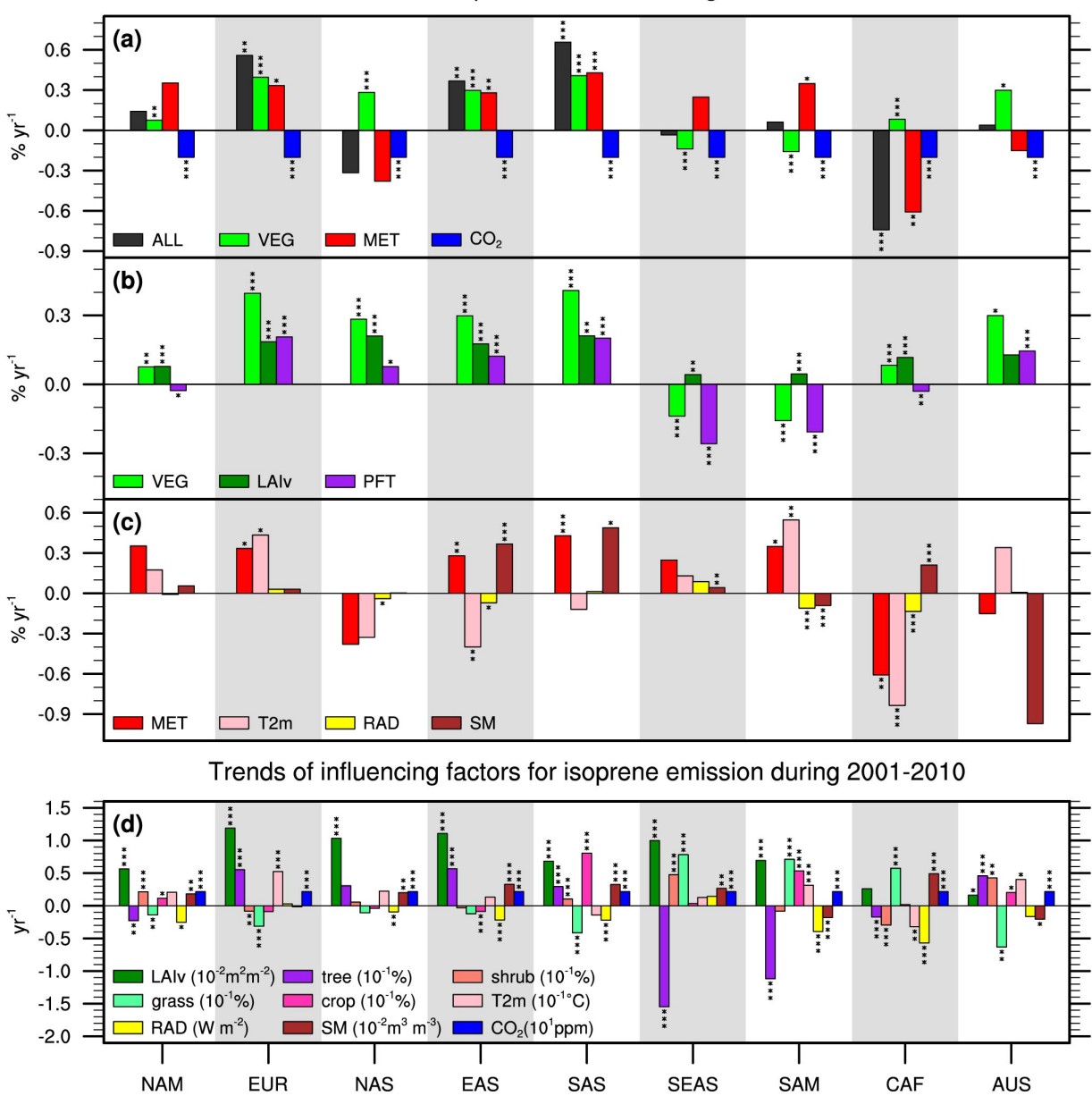

**Figure 9: Trends of isoprene emissions and associated influencing factors from 2001–2020 in nine regions. (a) ALL represents simulated results considering interannual variability of all drivers, while VEG, MET, and $CO_2$ represent simulated results considering only interannual variability of vegetation, meteorology, and $CO_2$ concentrations, respectively. (b) Contributions from individual vegetation parameters including LAIv (leaf area index of vegetation covered surfaces) and PFT (plant functional types). (c) Contributions from individual meteorological factors including T2m (surface 2 m temperature), RAD (surface solar radiation), and SM (soil moisture). (d) Trends of individual influencing factors including vegetation parameters, meteorological factors, and $CO_2$ concentrations. Tree, shrub, grass, and crop represent the cover for the four plant functional types (PFTs). The single, double, and triple asterisks denote 90%, 95%, and 99% confidence levels (CI) using the Mann-Kendall test, respectively.**

**4 Discussion**

In this study, we simulate the global BVOC emissions over the last 20 years using the latest version of the BVOC emission model MEGANv3.2 with time-varying vegetation and meteorological parameters, and $CO_2$ concentrations. We quantify the impacts of the various driving factors on the BVOC emission trends. Isoprene emissions are found to increase significantly in Europe, East Asia, and South Asia, with comparable contributions from vegetation and meteorological factors. Warming in Europe and wetting of soil in East Asia and South Asia lead to an increase in the isoprene emissions, while cooling in Central Africa leads to a significant decrease in the isoprene emissions.

The results presented here demonstrate the heterogeneous spatial and temporal variabilities of vegetation and meteorological factors, leading to different trends of BVOC emissions in different regions of the world. In some regions (e.g., Europe, East Asia, and South Asia), vegetation and meteorological factors combine to promote the BVOC emissions, resulting in significant positive trends, while in other regions, such as South America and Southeast Asia, the effects of vegetation and meteorological factors on the BVOC emissions offset each other, resulting in weak emission trends. For vegetation parameters, the biomass-related parameter LAIv and type-related parameter PFT cover are both proportional to the intensity of BVOC emissions. Although multiple satellite data reveal a global trend towards greening (i.e., increased LAI), there is a substantial decrease in PFT cover (i.e., tree cover) in some local regions of Southeast Asia and South America, leading to a decreasing trend of BVOC emissions there.

Note that the selection of the reference year (i.e., year 2001 in Table 1) may cause variations in simulated BVOC emissions, mostly affecting the magnitude rather than the sign of the absolute trends. Since this study focuses on the relative trends in BVOC emissions (i.e., ratio of absolute trend to multi-year means), differences in the reference year have little effect on the magnitude and sign of our estimation results. We show that during 2001-2020, global isoprene emissions decrease by a trend of $-0.07\%$ $yr^{-1}$[1], with changes in the meteorological factors, vegetation, and $CO_2$ concentrations contributing to a trend of $0.15\%$ $yr^{-1}$, $-0.03\%$ $yr^{-1}$, and $-0.20\%$ $yr^{-1}$, respectively. Some previous studies (e.g., Chen et al, 2018; Opacka et al, 2021; Sindelarova et al. 2022) also investigated the trends of isoprene emissions during 2000s and 2010s and obtained similar results about the positive trends due to meteorological changes and negative trends due to vegetation change. Chen et al. (2018) showed that the isoprene emission trend was $-0.1\%$ $yr^{-1}$ between 2000 and 2015. Sindelarova et al. (2022) used a time-varying ERA5 meteorological data and a static CLM4 land cover map to calculate the isoprene emission for 2000-2019 and suggested an isoprene emission trend of $0.35\%$ $yr^{-1}$. When replacing the static land cover map with the annually varying ESA-CCI data, Sindelarova et al. (2022) also found that the emission trend decreases to $0.24\%$ $yr^{-1}$. Opacka et al. (2021) found that the MEGAN-MOHYCAN model driven by the time-varying ERA-Interim meteorological field yielded an isoprene emission trend of $0.94\%$ $yr^{-1}$ for 2001-2016, while time-varying vegetation parameters can offset the positive trend by $0.04\%$ $yr^{-1}$ (based on MODIS land cover data) or $0.33\%$ $yr^{-1}$ (based on modified MODIS land cover data). The differences in the magnitude of trends due to meteorological changes or vegetation changes may be ascribed to the difference

in the input data (meteorology and vegetation) and emission parameterizations. In fact, driven by the ERA5, ERA-Interim, and MERRA-2 meteorological fields, the model simulates a trend of isoprene emission from 0.15% to 0.94% $yr^{-1}$ (a factor of 6 difference). Different vegetation data lead to the simulated isoprene emission trends differing from −0.03% to −0.33% $yr^{-1}$ (a factor of 11 difference). In addition, different model parameterizations can lead to large discrepancies in emission changes. For example, based on the same vegetation data, Li et al. (2022) showed that isoprene emission induced

by vegetation changes increased by about 20.6% over China between 2001 to 2018, which is larger than that in our study (12.0%). The difference can be explained by the fact that we use different vegetation emission factors and vegetation activity factors from Li et al. (2022).

       Vegetation cover tends to increase in a large part of global land areas and decrease in smaller areas such as South America (i.e., Amazon) and Southeast Asia (Fig. 6a). The trends of global isoprene emission due to vegetation cover change is largely

determined by the decreasing trends in Amazon and Southeast Asia due to the large contributions from these regions to global total isoprene emissions (Fig. 3a, Fig. 8f). The decrease of isoprene emissions in Amazon and Southeast Asia is mainly due to the change in PFT, such as from the original tropical broadleaf evergreen forest to broadleaf deciduous trees with agricultural economic benefits (Fig. 7, Fig. 9). In some areas, such as Europe, East Asia, and South Asia, the increased forest coverage due to afforestation contributes significantly to the increase in isoprene emissions, which is similar to the results from Chen et al.

(2018). However, Sindelarova et al. (2022) simulated little changes in isoprene emissions in Europe, mainly due to differences in the satellite PFT data used in their and our studies.

       In this study, we use the latest version of MEGAN model, which considers more refined factors compared to its predecessors. However, there are still large uncertainties in simulated BVOC emissions and their trends, such as in the treatment of drought and heat wave stress, emission factors of certain tree species, etc. Previous observations have shown that

isoprene and monoterpene emissions are affected differently by drought severity (Brilli et al., 2007; Kaser et al., 2022; Otu-Larbi et al., 2020; Simpraga et al., 2011). Isoprene emissions remain unchanged under mild drought, but increase under moderate drought with increased leaf temperature due to changes in stomatal conductance (Kaser et al., 2022). However, under severe drought, isoprene emissions drop due to reduced substrate supply (Brilli et al., 2007). The effects of drought on monoterpenes are similar to those of isoprene (Lavoir et al., 2009; Ormeno et al., 2007). Therefore, these processes need to be

better described in the model (Wang et al., 2022). In addition, more BVOC flux observations are urgently needed for model validations. In particular, the vegetation emission factors for tree species may be largely biased due to a scarcity of available observations and need to be further refined.

## 5 Conclusions

In this study, the time-varying meteorological data and satellite observations are used to drive the latest BVOC emission model MEGANv3.2 to simulate global BVOC emissions for the past 20 years. The contributions from different factors to the trends of BVOC emissions from global to regional scales are quantified.

Compared with site observations, the model can simulate isoprene emissions within a factor of ten at most stations but systematically overestimates the monoterpene emissions. Compared to space-borne isoprene retrievals, the model can capture high isoprene emission regions such as South America and Central Africa. There are large seasonal variations in isoprene emissions, which are mainly determined by temperature and vegetation variations. The relative contribution of isoprene (monoterpene) emissions to BVOC emissions tends to decrease (increase) with latitudes, which is mainly ascribed to the meridional variations of PFT cover and corresponding emission factors.

Isoprene emissions increase significantly in Europe, East Asia, and South Asia (at rates of $0.37 - 0.66\%$ $yr^{-1}$), with changes in both vegetation and meteorological factors contributing almost equally to the trends. For different meteorological factors, isoprene emission trends are mainly driven by the increase in temperature in Europe and by the increase in soil moisture in East and South Asia. In South America and Southeast Asia, shifts in PFT cover leads to a significant decrease in the BVOC emissions, which cancels out nearly half of the increasing trends induced by the changes in meteorological parameters. In addition, despite the increase in global mean temperature, there is a decrease in temperature in Central Africa, resulting in a significantly decrease trend in isoprene emission in this region ($-0.74\%$ $yr^{-1}$). The dominant factors of monoterpene emission trends are similar to those of isoprene emissions, while monoterpene emissions show a stronger increasing trend or a weaker decreasing trend in most regions. In addition, monoterpene emissions are more sensitive to changes in LAIv, resulting in more pronounced increasing trends in greening hotspots.

Overall, our study highlights the significant BVOC emission trends both globally and regionally. More importantly, the results from this study clarify the contributions from different drivers and deepen our understanding of long-term BVOC emission trends at regional to global scales. Changes of BVOC emissions may have important impacts on ozone and atmospheric particle formation, which consequently impact the atmospheric chemistry, radiation and climate. These interactions involving BVOCs will be investigated by using a coupled meteorology-chemistry with the BVOC emission model.

**Code Availability**

The MEGANv3.2 source code is available at https://bai.ess.uci.edu/megan/data-and-code (last access: 21 Nov 2022).

## Data Availability

The model output of MEGANv3.2 is archived at NCAR Cheyenne supercomputer and will be available one the manuscript is published online. The LAI data from Yuan et al. (2011) are downloaded from http://globalchange.bnu.edu.cn /research/laiv6 (last access: 21 Nov 2022). The MODIS MCD12C1 land cover product Version 6 and MODIS MOD44B VCF Version 6 datasets are available on the website of the Land Processes Distributed Active Archive Center (LP DAAC) at https://lpdaac.usgs.gov/dataset_discovery/modis/modis_products_table (last access: 21 Nov 2022). The MERRA-2 data are obtained from https://goldsmr4.gesdisc.eosdis.nasa.gov/data/MERRA2/ (last access: 21 Nov 2022). The globally averaged $CO_2$ concentration data are obtained from https://gml.noaa.gov/ccgg/trends/gl_data.html (last access: 21 Nov 2022). The IASB-TD-OMI data are obtained from http://www.globemission.eu/ (last access: 21 Nov 2022). The isoprene columns data used in this work are available at https://doi.org/10.13020/v959-dr15. (last access: 21 Nov 2022).

## Author Contributions

XL, CW, and HW planned and organized the project. HW designed the experiments and performed the simulations with help and input from XL, CW, and GL. HW led the analysis and wrote the first draft of the paper. All co-authors participated in discussions on data analysis and revised the paper.

## Competing Interests

At least one of the (co-)authors is a member of the editorial board of Atmospheric Chemistry and Physics.

## Acknowledgement

This study was supported by the National Key Research and Development Program of China (grant 2020YFA0607801) and the National Natural Science Foundation of China (grant 41830966, 41975119, and 42275086). We thank the global MEGAN community for their tireless work to improve the model. We would like to acknowledge the use of computational resources for conducting the model simulations at the NCAR-Wyoming Supercomputing Center provided by the NSF and the State of Wyoming and supported by NCAR's Computational and Information Systems Laboratory. This work was also supported by the National Key Scientific and Technological Infrastructure project "Earth System Science Numerical Simulator Facility" (EarthLab). The authors thank the MERRA-2, MODIS, and other teams for the data used in this study.

**Financial Support**

This study was supported by the National Key Research and Development Program of China (grant 2020YFA0607801) and the National Natural Science Foundation of China (grant 41830966, 41975119, and 42275086).

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
