# Peer review of "Regional to global distributions, trends, and drivers of biogenic volatile organic compound emission from 2001 to 2020"

_EGUsphere, 2023_

## Author Comment (AC1)

**Responses to the Reviewers' Comments**

We would like to thank the editor and two anonymous reviewers for their helpful comments and suggestions. All authors have read the revised manuscript and agreed with the submission in its revised form.

Reviewers' comments are in black color, our responses are in blue color, and our corresponding revisions in the manuscript are in red color.

**Response to the Reviewer #1**

General Comments:

BVOCs are important precursors to ozone and secondary organic aerosols in the atmosphere. In this manuscript, the authors focused on a comprehensive analysis of trends in BVOC emissions from 2001-2020 on a regional to global scales and identified the contribution of various driving factors to these trends. The manuscript is well written, and I suggested the acceptance after addressing the comments below.

**Response:** We thank the reviewer for the encouraging comments. We have revised the manuscript following your comments.

1. The authors used a newer version of MEGAN. Could the authors elaborate the major advancement compared to previous versions such as MEGAN 2.1?

**Response:** Thank you. Following your comment, we have added the following text for the major advancement in the new MEGANv3.2 compared to MEGANv2.1:

in Section 2.1 "MEGANv3.2":

"Specifically, while MEGANv2.1 uses a look-up table of emission factors for the 15 PFTs corresponding to the biological emission classes (see Table 2 in Guenther et al. (2012)), MEGANv3.2 uses the so-called Emission Factor Processor, to estimate the landscape average emission factors, which are based on the following three databases: (1) Growth form datasets for four PFTs: tree, shrub, grass, and crops; (2) Ecotype datasets: composed of a mix of emission-specific tree species/grass associated with specific emission capacities; and (3) Updated tree species/grass datasets corresponding to the biogenic emission classes. These updates can distinguish the differences in vegetation emission factors in regions with the same PFT but with varying plant species. The new version also considers the additional stress factors of emissions by using the simple threshold function, including high/low temperature, strong wind, and heavy $O_3$ pollution."

2. Are the observations only in 2013? The seasonal variations of isoprene flux in MEGAN appears to be very small, which seems to be quite different from the observations. Any explanations?

**Response:** Yes, the observations are only in 2013. The magnitude of seasonal variations of the modeled isoprene flux is small compared to the observations. We have added the explanations for the small seasonal variations in simulated isoprene flux in Section 3.1.1 "Spatial distribution of BVOC emissions":

"The large bias in the seasonal variation of isoprene fluxes in MEGANv3.2 may be due to a lack of representation of the isoprene emission capacity of tree species at different leaf ages (Alves et al., 2018). Additionally, the model bias arises from a lack of realistic representations of leaf phenology, canopy structure, soil moisture feedbacks, and variation in isoprene emissions due to the complex biodiversity in the Amazon region."

3.    Line 274: These discrepancies are mainly ascribed to the differences in vegetation emission factors between the two versions of MEGAN. Could the authors add some explanations the emission factors from which vegetation are more accurate?

**Response:** Thank you for the comment. We have added some explanation and discussion of the accuracy of the vegetation emission factors for MEGANv3.2 and MEGANv2.1 in Section 3.1.1 "Spatial distribution of BVOC emissions":

"Note that MEGANv2.1 only utilizes fixed emission factors corresponding to the PFTs, but the PFT is insufficient to characterize the emission factors, e.g., tree species with the same PFT may have very different BVOC emission rates. MEGANv3.2 further considers differences in emission factors for tree species with the same PFTs. Thus, the vegetation emission factors in MEGANv3.2 are more accurately represented. However, we note that the uncertainties associated with emission factors are still large due to the limited observational data (Guenther et al., 2020)."

4.    Line 367: Could the authors explain why for monoterpene emissions, only the effects of vegetation and meteorological factors are considered?

**Response:** Thank for your comment. We have added the reasons why for monoterpene emissions in the model only the effects of vegetation and meteorological factors are considered in Section 3.2 "Trends of BVOC emissions":

"Different from isoprene emissions, there is no statistically significant effect of $CO_2$ concentration on monoterpene emissions as suggested by previous studies (Malik et al., 2019, 2023). Therefore, monoterpene emissions in MEGANv3.2 only consider the effects of vegetation and meteorological factors and show a significantly positive trend of 0.34% $yr^{-1}$ globally (Fig. S2)."

**References**

Alves, E. G., Tóta, J., Turnipseed, A., Guenther, A. B., Vega Bustillos, J. O. W., Santana, R. A., Cirino, G. G., Tavares, J. V., Lopes, A. P., Nelson, B. W., de Souza, R. A., Gu, D., Stavrakou, T., Adams, D. K., Wu, J., Saleska, S., and Manzi, A. O.: Leaf phenology as one important driver of seasonal changes in isoprene emissions in central Amazonia, Biogeosciences, 15, 4019-4032, 10.5194/bg-15-4019-2018, 2018.

Guenther, A., Jiang, X., Shah, T., Huang, L., Kemball-Cook, S., and Yarwood, G.: Model of Emissions of Gases and Aerosol from Nature Version 3 (MEGAN3) for Estimating Biogenic Emissions, in: Springer Proceedings in Complexity, Springer, Cham, 187–192, https://doi.org/10.1007/978-3-030-22055-6_29, 2020.

Guenther, A. B., Jiang, X., Heald, C. L., Sakulyanontvittaya, T., Duhl, T., Emmons, L. K., and Wang, X.: The Model of Emissions of Gases and Aerosols from Nature version 2.1 (MEGAN2.1): an extended and updated framework for modeling biogenic emissions, Geosci. Model Dev., 5, 1471-1492, 10.5194/gmd-5-1471-2012, 2012.

Malik, T. G., Gajbhiye, T., and Pandey, S. K.: Some insights into composition and monoterpene emission rates from selected dominant tropical tree species of Central India: Plant-specific seasonal variations, Ecological Research, 34, 821-834, 10.1111/1440-1703.12058, 2019.

Malik, T. G., Sahu, L. K., Gupta, M., Mir, B. A., Gajbhiye, T., Dubey, R., Clavijo McCormick, A., and Pandey, S. K.: Environmental Factors Affecting Monoterpene Emissions from Terrestrial Vegetation, Plants, 12, 3146, https://doi.org/10.3390/plants12173146, 2023.

---

## Author Comment (AC2)

**Responses to the Reviewers' Comments**

We would like to thank the editor and two anonymous reviewers for their helpful comments and suggestions. All authors have read the revised manuscript and agreed with the submission in its revised form.

Reviewers' comments are in black color, our responses are in blue color, and our corresponding revisions in the manuscript are in red color.

**Response to the Reviewer #2**

General Comments:

This is a study of great relevance to the atmospheric pollution modeling community. The trends presented and the sensitivity of MEGAN 3.2 to various parameters and its geographical distribution are of great interest to all of us who model both emissions and air quality. It provides interesting insights into how BVOC emissions can change at specific regions and highlights the significant impact of land-use changes and global warming.

**Response:** We thank the reviewer for the encouraging comments.

I would like to ask the authors if they have reviewed or analyzed the uncertainty associated with the new emission factors used in MEGAN. Have they utilized those with the highest confidence index (denoted as 'J' in the code)? Or do they consider it worthwhile to address these factors or their spatial distribution in the future? Are these emission factors the default values in MEGAN for tree, shrub, grass, and crop categories?

**Response:** Thank you for your comments and questions. In fact, MEGANv3.2 provides an open-source and expandable database of species-specific emission factors. The model code uses these emission factors by default, and there is no option to set a confidence index (denoted as 'J' in the code). Therefore, in this study, we did not carry out sensitivity analyses directly for the uncertainty associated with the new emission factors, but instead used ground- and satellite-based observations, and previous simulation results to evaluate our modeled BVOC emission fluxes.

Currently, most of the tree emission factors provided by MEGANv3.2 come from observations in the United States as well as from numerous literature data. Therefore, there is an urgent need to conduct observations in other hotspot regions around the globe in the future. This point has been emphasized in the Discussion section.

These emission factors are not the default values in MEGAN for tree, shrub, grass, and crop categories. Compared to MEGANv2.1, MEGANv3.2 can distinguish the differences in vegetation emission factors in regions with the same PFTs but with varying plant species. We have added a detailed explanation of the MEGANv3.2 emission factor methodology in the model introduction section.

Text about the MEGANv3.2 emission factor calculation methodology has been added in Section 2.1 "MEGANv3.2":

"Specifically, while MEGANv2.1 uses a look-up table of emission factors for the 15 PFTs corresponding to the biological emission classes (see Table 2 in Guenther et al. (2012)), MEGANv3.2 uses the so-called Emission Factor Processor, to estimate the landscape average emission factors, which are based on the following three databases: (1) Growth form datasets for four PFTs: tree, shrub, grass, and crops; (2) Ecotype datasets: composed of a mix of emission-specific tree species/grass associated with specific emission capacities; and (3) Updated tree species/grass datasets corresponding to the biogenic emission classes. These updates can distinguish the differences in vegetation emission factors in regions with the same PFT but with varying plant species. The new version also considers the additional stress factors of emissions by using the simple threshold function, including high/low temperature, strong wind, and heavy O3 pollution."

Was there any anomaly in the reference year of 2001 that could potentially bias the study in specific regions? Did the authors find any unexpected anomalies that they did not anticipate?

**Response:** Thank you for your comments. Since there are significant inter-annual variations in the drivers (vegetation, meteorology, and $CO_2$) affecting BVOC emissions in some specific regions. The selection of the reference year may lead to differences in the modeled BVOC emissions, primarily affecting the magnitude rather than the sign of the absolute trends. However, this study focuses on the relative trends in BVOC emissions. Differences in the reference year have little effect on the magnitude and sign of our estimation results.

We have added the following discussion of the impact of the reference year selection on our results in Section 4 "Discussion":

"Note that the selection of the reference year (i.e., year 2001 in Table 1) may cause variations in simulated BVOC emissions, mostly affecting the magnitude rather than the sign of the absolute trends. Since this study focuses on the relative trends in BVOC emissions (i.e., ratio of absolute trend to multi-year means), differences in the reference year have little effect on the magnitude and sign of our estimation results."

Possible issues I have detected:

- In line 70, based on my reading of the rest of the manuscript, shouldn't the range '0.04-0.33% $yr^{-1}$' be negative?

**Response:** Sorry for the ambiguity of this sentence, we've corrected it.

"… and pointed out that land cover changes from 2001 to 2016 mitigate the isoprene emissions ranging from -0.33% to -0.04% $yr^{-1}$"

- Please review the use of capitalization for the acronyms 'LAI' and 'VCF' in both the text and figure captions.

  **Response:** Thank you for your comment. We have reviewed and corrected the use of capitalization for the acronyms 'LAI' and 'VCF' in both the text and figure captions.

- In line 545, it should be corrected with "activity factors".

  **Response:** Done. Thank you.

**References**

Guenther, A. B., Jiang, X., Heald, C. L., Sakulyanontvittaya, T., Duhl, T., Emmons, L. K., and Wang, X.: The Model of Emissions of Gases and Aerosols from Nature version 2.1 (MEGAN2.1): an extended and updated framework for modeling biogenic emissions, Geosci. Model Dev., 5, 1471-1492, 10.5194/gmd-5-1471-2012, 2012.

---

## Author Response (AR2)

**Responses to the Reviewers' Comments**

We would like to thank the editor and two anonymous reviewers for their helpful comments and suggestions. All authors have read the revised manuscript and agreed with the submission in its revised form.

Reviewers' comments are in black color, our responses are in blue color, and our corresponding revisions in the manuscript are in red color.

**Response to the Reviewer #3**

This work provides a comprehensive analysis of the different trends impacting BVOC emissions at global and at regional scale. Wang et al. use the state-of-the-art emission model MEGANv3.2 to evaluate BVOC emission trends by varying specific drivers, and keeping other variables fixed in time. These simulations provide an important insight into the importance of these variables, such as temperature changes, land cover changes, and $CO_2$ concentration changes in the past two decades.

The paper is well written and features an array of interesting figures and graphs. I would recommend this paper to be published after minor revisions.

**Response:** We thank the reviewer for the encouraging comments. We have revised the manuscript following your comments.

General Comments:

- It should be noted that there are large uncertainties in models of bottom-up inventories, such as in the treatment of drought stress, emission factors of certain tree species, etc. Although MEGAN3 provides an improved model of some important emission drivers, an in-depth discussion on model uncertainties is somewhat lacking.

**Response:** Thank you for your comments. We have added an in-depth discussion on model uncertainties (such as in the treatment of drought stress, emission factors of certain tree species, etc.) in Section 4 "Discussion":

"In this study, we use the latest version of MEGAN model, which considers more refined factors compared to its predecessors. However, there are still large uncertainties in simulated BVOC emissions and their trends, such as in the treatment of drought and heat wave stress, emission factors of certain tree species, etc. Previous observations have shown that isoprene and monoterpene emissions are affected differently by drought severity (Brilli et al., 2007; Kaser et al., 2022; Otu-Larbi et al., 2020; Simpraga et al., 2011). Isoprene emissions remain unchanged under mild drought, but increase under moderate drought with increased leaf temperature due to changes in stomatal conductance (Kaser et al., 2022). However, under severe drought, isoprene emissions drop due to reduced substrate supply (Brilli et al., 2007). The effects of drought on monoterpenes are similar to those of isoprene (Lavoir et al., 2009; Ormeno et al., 2007).

Therefore, these processes need to be better described in the model (Wang et al., 2022). In addition, more BVOC flux observations are urgently needed for model validations. In particular, the vegetation emission factors for tree species may be largely biased due to a scarcity of available observations and need to be further refined."

- Introduction: I think it would be good to introduce the main BVOC compounds in the introduction (i.e. isoprene, monoterpenes, methanol, …).

**Response:** Thank you for your comments. We have introduced the main BVOC compounds in the introduction.

"Isoprene and monoterpenes (e.g., α-pinene, β-pinene, limonene) are the most prevalent BVOC species, and other species include sesquiterpenes, methanol, ethanol, etc."

- Section 2.2.1: It is not entirely clear how the vegetation dataset was constructed. The authors write that the high-resolution MODIS vegetation map was mapped onto the four PFTs of MEGANv3.2. I presume the interpolation is used to calculate the growth form fractions used by the Emission Factor Processor in Eq. 2? Furthermore, it is unclear to me where the ecotype dataset mentioned in Sect. 2.1 comes from.

**Response:** Sorry for the ambiguity. The high-resolution MODIS vegetation map was mapped onto the four PFTs of MEGANv3.2, which is used to calculate the growth form fractions used by the Emission Factor Processor in Eq. 2. In addition, we have added a description of the ecotype dataset sources and have further clarified how the vegetation data were created in section 2.2.1"Vegetation datasets":

"To calculate the growth form fractions used by the Emission Factor Processor in Equation 2, the selected 17 MODIS IGBP (International Geosphere Biosphere Programme) global vegetation classification types from above MODIS PFT product were mapped to four main PFT classification types (i.e., tree, shrub, grass, and crop) in MEGANv3.2 based on methods from Sulla-Menashe and Friedl (2018). The ecotype dataset mentioned in Section 2.1 is based on satellite imagery and ground surveys and comes from the MEGAN model development group (https://bai.ess.uci.edu/megan/data-and-code/growth-form-and-ecotypes, last access: 21 Nov 2022)"

- Section 3.1.1: The comparison of in situ isoprene measurements with the model is not always representative. Isoprene has a very short lifetime in the atmosphere, which implies that its measured fluxes depend on the local tree or plant species near the observation site. Are the assigned vegetation types shown in Fig. 1 based on the local vegetation of the observation site? And how do they differ from the average PFT derived from Sect. 2.2.1?.

**Response:** Thank you for your comments and questions. The assigned vegetation types shown in Fig. 1 based on the local vegetation of the observation site. Comparing the site PFTs to the average four PFTs derived from Sect. 2.2.1 (Fig. R1), we find that in areas where a single PFT dominates, the corresponding site PFTs are representative. However, in areas with multiple PFTs, the corresponding site PFTs are weakly

representative. We have added an analysis of site PFT representativeness in Section 3.1.1"Spatial distribution of BVOC emissions":

[Figure]

**Figure R1: (a-d) Cover fractions of the four plant functional types (PFTs) and (e) the location distribution of isoprene and monoterpene observation sites based on the literature collection (the five-pointed star represents the location of the K34 tower site, EBF: evergreen broadleaf forest, DBF: deciduous broadleaf forest, ENF: evergreen needleleaf forest, Grass: grassland).**

"In addition, note that the comparison of in situ isoprene measurements with the model is not always representative. Isoprene has a very short lifetime (minutes to hours) in the atmosphere, which implies that its measured fluxes depend on the local tree or plant species near the observation site."

- Figure 9: Do the simulations of the individual drivers refer to the simulations in Table 1? The trends are described as contributions of the individual meteorological factors in panel 9c, but from Table 1 I understand that only one of the drivers is fixed and the two others are allowed to vary over the simulations. Consequently, if the T2m simulation implies T2m is fixed, why is the trend large for Central Africa and Europe? And the same goes for soil moisture and solar radiation trends.

**Response:** Sorry for the ambiguity. The simulations of the individual drivers refer to the simulations in Table 1, but the individual driver contribution is obtained from the difference between the results of the two experiments. Details are given in section 2.3.

Specifically, the difference between EMIT_VEG and EMIT_VEG_FIX_PFT/EMIT_VEG_FIX_LAIv represents the contribution of LAIv and PFT historical changes to the BVOC emission trends. The difference between EMIT_MET and EMIT_MET_FIX_T2m, EMIT_MET_FIX_RAD, and EMIT_MET_FIX_SM represent the impact of temperature, light, and soil moisture changes to the BVOC emission trends, respectively. We have added a clarification of the experiments in section 3.3.2 "Contribution of drivers to BVOC emission trends":

"The vegetation parameters can be decomposed into LAIv and PFT cover (Fig. 9b, Equations 2 and 3; the difference between EMIT_VEG and EMIT_VEG_FIX_PFT/EMIT_VEG_FIX_LAIv in Table 1)"

"Three meteorological variables namely temperature, radiation, and soil moisture, which are the main influencing factors from meteorology, are selected for quantifying their effects on the trends of isoprene emissions (Fig. 9c, d; the difference between EMIT_MET and EMIT_MET_FIX_T2m, EMIT_MET_FIX_RAD, and EMIT_MET_FIX_SM in Table 1)."

Specific comments:

- Line 38: This sentence is unclear, as BVOC emissions are almost entirely determined by vegetation and meteorology, whereas $CO_2$ concentrations play a much smaller, secondary effect due to its homogeneous distribution.

**Response:** Agreed, thanks for the comment, we've fixed it.

"The BVOC emissions are determined by many environmental factors such as vegetation, meteorology, and carbon dioxide ($CO_2$) concentrations."

- Line 44: Add reference for the $CO_2$ inhibition effect. Why does it only impact isoprene?

**Response:** Thank you for your comments and questions. We have added references for the $CO_2$ inhibition effect. The $CO_2$ concentration not only affects isoprene emissions, but may also affect other BVOC components such as monoterpenes. However, the effects are more complex and are not currently introduced in the MEGAN model. We have revised the sentence.

"…elevated $CO_2$ can suppress the emissions of the major BVOC component (e.g., isoprene) (Heald et al.,2009; Wilkinson et al., 2009)."

- Line 240-243: The authors attribute the potential reasons for the differences between their study and IASB-TD-OMI to different model parameters and different meteorological datasets used. Although these factors indeed influence the top-down emissions fluxes, they are mainly constrained by formaldehyde column measurements. Consequently, the reasons for the difference are more complex, as these are two very different methods.

**Response:** Thank you for your comments, we have added the relevant explanations

about estimation methods in Section 3.1.1"Spatial distribution of BVOC emissions":

"The differences are mainly concentrated in South America, Central Africa, and Southeast Asia, which may be partly due to differences in estimation methods (i.e., top-down vs. bottom-up), emission model parameters (e.g., vegetation emission factors) and the meteorological datasets used (MERRA-2 vs. ERA-Interim)."

- Line 289: Which new BVOC components were added to MEGANv3.2? Does any on these new components contribute significantly to the total of 279.6 Tg yr$^{-1}$?

**Response:** Thank you for your questions. Compared to MEGANv2.1, MEGANv3.2 mainly adds more "stress BVOC" components, i.e. the number of "stress BVOC" components has been expanded from 15 to 55. New species include: 1-pentanol, penten, butanenitrile, 3-metfuran, 3-metthiophene, 3-pentanone, 8-heptadecene, acetophenone, benzene, benzyl benzoate, chavicol, cinnamaldehyde, cinnamic acid, etc. In addition, butane has also been added. The details can be found in the file "SPC_NOCONVER.EXT" in the MEGANv3.2 model (https://bai.ess.uci.edu/megan/data-and-code/megan32). In our current experiments, the model outputs the newly added species in a single category as the "stress BVOC" component. This means that it is not enough to isolate the contribution of the new species, which could be of interest for future studies. We have added a description of the new BVOC components in Section 3.1.1"Spatial distribution of BVOC emissions":

"which can be partly attributed to more BVOC components (e.g., butane, butanenitrile, acetophenone, benzene cinnamaldehyde, cinnamic acid, etc.) considered in MEGANv3.2 than in MEGANv2.1."

- Line 321: The study of Sindelarova et al. (2014) shows a peak in October, not December.

**Response:** Thank you for your meticulous checking. We've recalculated the relevant emission inventories and found that you're right. We've fixed it!

- Line 368: Technically, the increase of $CO_2$ concentration is also partly responsible for global warming and thus higher temperatures (see e.g. Monson et al. 2007). Additionally, increased $CO_2$ concentration can potentially lead to a larger LAIv (also Monson et al. 2007). Consequently, this sentence could be rephrased to specifically the $CO_2$ inhibition effect resulting in the decrease of -0.20% yr$^{-1}$ of isoprene emissions.

**Response:** Thank you for your comments. We have rephrased the relevant sentences in Section 3.1.1"Trends of BVOC emissions":

"As expected, the $CO_2$ inhibition effect results in a significant decrease of -0.20% yr$^{-1}$ in isoprene emissions (Fig. 5). Note that this study uses a globally uniform and yearly mean $CO_2$ concentration without considering spatial and seasonal variations of $CO_2$ concentration. Additionally, the $CO_2$ concentration can also indirectly affect isoprene emissions by changing meteorological and vegetation factors. Specifically, the increase of $CO_2$ concentration is partly responsible for global warming and thus higher temperatures (e.g. Monson et al., 2007), and the increased $CO_2$ concentration can

potentially lead to a larger LAIv (Monson et al., 2007). These indirect effects are not explicitly considered in this study."

References:

Monson, R. K., et al. (2007), Isoprene emission from terrestrial eco systems in response to global change: Minding the gap between models and observations, Philos. Trans. R. Soc. A,365, 1677–1695, doi:10.1098/rsta.2007.2038.

- Line 371: In Malik et al. (2023), the authors also argue that evidence for the effect of soil moisture stress on monoterpene emissions is inconclusive. Is there a reason why this effect is included for monoterpenes and not for $CO_2$ inhibition?

**Response:** Thank you for your questions. In fact, the effect of monoterpenes on drought is complex and may depend on the drought intensity, as well as the drought-tolerance of the plant species. Previous study has been found that monoterpene emissions increase during mild drought while decrease during severe drought (Ormeno et al., 2007). In this study, although we use the latest version of MEGAN model, which considers more refined factors compared to its predecessors, the parameterization of soil moisture stress on BVOC emissions are still relatively simple in the model and more research is needed to explore this in depth. We have added some discussion of the soil moisture stress on BVOC emissions in Section 4 "Discussion":

"However, there are still large uncertainties in simulated BVOC emissions and their trends, such as in the treatment of drought and heat wave stress, emission factors of certain tree species, etc. Previous observations have shown that isoprene and monoterpene emissions are affected differently by drought severity (Brilli et al., 2007; Kaser et al., 2022; Otu-Larbi et al., 2020; Simpraga et al., 2011). Isoprene emissions remain unchanged under mild drought, but increase under moderate drought with increased leaf temperature due to changes in stomatal conductance (Kaser et al., 2022). However, under severe drought, isoprene emissions drop due to reduced substrate supply (Brilli et al., 2007). The effects of drought on monoterpenes are similar to those of isoprene (Lavoir et al., 2009; Ormeno et al., 2007). Therefore, these processes need to be better described in the model (Wang et al., 2022)."

- The title of Sect. 3.3 is the same as of Sect. 3.3.2.

**Response:** Thank you for your comments, we've revised title of Sect. 3.3.2 to "Contribution of drivers to BVOC emission trends".

Lay-out comments: write percentage changes in mathematical format to get a better minus-sign and to avoid splitting at the end of a line.

**Response:** Done, thank you.

**References:**

Brilli, F., Barta, C., Fortunati, A., Lerdau, M., Loreto, F., and Centritto, M.: Response of isoprene emission and carbon metabolism to drought in white poplar (Populus alba) saplings, New Phytologist, 175, 244-254, https://doi.org/10.1111/j.1469-8137.2007.02094.x, 2007.

Heald, C. L., Wilkinson, M. J., Monson, R. K., Alo, C. A., Wang, G., and Guenther, A.: Response of isoprene emission to ambient $CO_2$ changes and implications for global budgets, Global Change Biology, 15, 1127-1140, https://doi.org/10.1111/j.1365-2486.2008.01802.x, 2009.

Kaser, L., Peron, A., Graus, M., Striednig, M., Wohlfahrt, G., Juráň, S., and Karl, T.: Interannual variability of terpenoid emissions in an alpine city, Atmos. Chem. Phys., 22, 5603-5618, 10.5194/acp-22-5603-2022, 2022.

Lavoir, A. V., Staudt, M., Schnitzler, J. P., Landais, D., Massol, F., Rocheteau, A., Rodriguez, R., Zimmer, I., and Rambal, S.: Drought reduced monoterpene emissions from the evergreen Mediterranean oak Quercus ilex: results from a throughfall displacement experiment, Biogeosciences, 6, 1167-1180, 10.5194/bg-6-1167-2009, 2009.

Monson, R. K., Trahan, N., Rosenstiel, T. N., Veres, P., Moore, D., Wilkinson, M., Norby, R. J., Volder, A., Tjoelker, M. G., Briske, D. D., Karnosky, D. F., and Fall, R.: Isoprene emission from terrestrial ecosystems in response to global change: minding the gap between models and observations, Philosophical Transactions of the Royal Society A: Mathematical, Physical and Engineering Sciences, 365, 1677-1695, doi:10.1098/rsta.2007.2038, 2007.

Ormeño, E., Mévy, J. P., Vila, B., Bousquet-Mélou, A., Greff, S., Bonin, G., and Fernandez, C.: Water deficit stress induces different monoterpene and sesquiterpene emission changes in Mediterranean species. Relationship between terpene emissions and plant water potential, Chemosphere, 67, 276-284, https://doi.org/10.1016/j.chemosphere.2006.10.029, 2007.

Otu-Larbi, F., Bolas, C. G., Ferracci, V., Staniaszek, Z., Jones, R. L., Malhi, Y., Harris, N. R. P., Wild, O., and Ashworth, K.: Modelling the effect of the 2018 summer heatwave and drought on isoprene emissions in a UK woodland, Global Change Biology, 26, 2320-2335, https://doi.org/10.1111/gcb.14963, 2020.

Šimpraga, M., Verbeeck, H., Demarcke, M., Joó, É., Pokorska, O., Amelynck, C., Schoon, N., Dewulf, J., Van Langenhove, H., Heinesch, B., Aubinet, M., Laffineur, Q., Müller, J. F., and Steppe, K.: Clear link between drought stress, photosynthesis and biogenic volatile organic compounds in Fagus sylvatica L, Atmospheric Environment, 45, 5254-5259, https://doi.org/10.1016/j.atmosenv.2011.06.075, 2011.

Wang, H., Lu, X., Seco, R., Stavrakou, T., Karl, T., Jiang, X., Gu, L., and Guenther, A. B.: Modeling Isoprene Emission Response to Drought and Heatwaves Within MEGAN Using Evapotranspiration Data and by Coupling With the Community Land Model, Journal of Advances in Modeling Earth Systems, 14, e2022MS003174, https://doi.org/10.1029/2022MS003174, 2022.

Wilkinson, M. J., Monson, R. K., Trahan, N., Lee, S., Brown, E., Jackson, R. B., Polley, H. W., Fay, P. A., and Fall, R.: Leaf isoprene emission rate as a function of atmospheric $CO_2$ concentration, Global Change Biology, 15, 1189-1200, https://doi.org/10.1111/j.1365-2486.2008.01803.x, 2009.